# Mechanically enhanced biogenesis of gut spheroids with instability-driven morphomechanics

Feng Lin[1,2,7], Xia Li[1,7], Shiyu Sun[1,3,7], Zhongyi Li[1,7], Chenglin Lv[1], Jianbo Bai[1], Lin Song[4], Yizhao Han[1], Bo Li [1] ✉, Jianping Fu [3,5,6] & Yue Shao [1,4] ✉

Region-specific gut spheroids are precursors for gastrointestinal and pulmonary organoids that hold great promise for fundamental studies and translations. However, efficient production of gut spheroids remains challenging due to a lack of control and mechanistic understanding of gut spheroid morphogenesis. Here, we report an efficient biomaterial system, termed micropatterned gut spheroid generator (μGSG), to generate gut spheroids from human pluripotent stem cells through mechanically enhanced tissue morphogenesis. We show that μGSG enhances the biogenesis of gut spheroids independent of micropattern shape and size; instead, mechanically enforced cell multilayering and crowding is demonstrated as a general, geometry-insensitive mechanism that is necessary and sufficient for promoting spheroid formation. Combining experimental findings and an active-phase-field morphomechanics theory, our study further reveals an instability-driven mechanism and a mechanosensitive phase diagram governing spheroid pearling and fission in μGSG. This work unveils mechanobiological paradigms based on tissue architecture and surface tension for controlling tissue morphogenesis and advancing organoid technology.

During embryogenesis, the gut tube gives rise to multiple vital organs, e.g., the esophagus, stomach, intestine, and lung, in a region-specific manner (Fig. 1a)[1,2]. By recapitulating such developmental processes, region-specific gut spheroids have been developed from human pluripotent stem cells (hPSCs) in vitro (Fig. 1b), which can be further differentiated to form organoids that model the development of various gastrointestinal and pulmonary organs[3–6]. These stem cell-based models not only are useful for elucidating the developmental programs underlying human organogenesis, but also hold promise for modeling diseases and advancing regenerative therapeutics[3,7–12]. Despite such fundamental and translational significance, scalable biomanufacturing of gut spheroids, which are precursors of gastrointestinal and pulmonary organoids[3–6,13–15], is currently limited by suboptimal efficiency using conventional monolayer-based induction[16], as well as a lack of mechanistic understanding of gut spheroid morphogenesis.

Micropatterns, which provide geometrically defined niches to cells cultured on 2D surfaces, have been recently demonstrated as powerful tools to enable the efficient generation and mechanistic investigation of multiple hPSC-based developmental models[17–22]. However, the applicability of micropatterns to enhance the biogenesis of gut spheroids, which requires morphogenic paradigms

[1]Institute of Biomechanics and Medical Engineering, Department of Engineering Mechanics, School of Aerospace Engineering, Tsinghua University, Beijing 100084, China. [2]Wenzhou Institute, University of Chinese Academy of Sciences, Wenzhou, Zhejiang 325000, China. [3]Department of Mechanical Engineering, University of Michigan, Ann Arbor, MI 48109, USA. [4]State Key Laboratory of Primate Biomedical Research, Institute of Primate Translational Medicine, Kunming University of Science and Technology, Kunming, Yunnan 650500, China. [5]Department of Biomedical Engineering, University of Michigan, Ann Arbor, MI 48109, USA. [6]Department of Cell & Developmental Biology, University of Michigan Medical School, Ann Arbor, MI 48109, USA. [7]These authors contributed equally: Feng Lin, Xia Li, Shiyu Sun, Zhongyi Li. ✉e-mail: libome@tsinghua.edu.cn; yshao@tsinghua.edu.cn

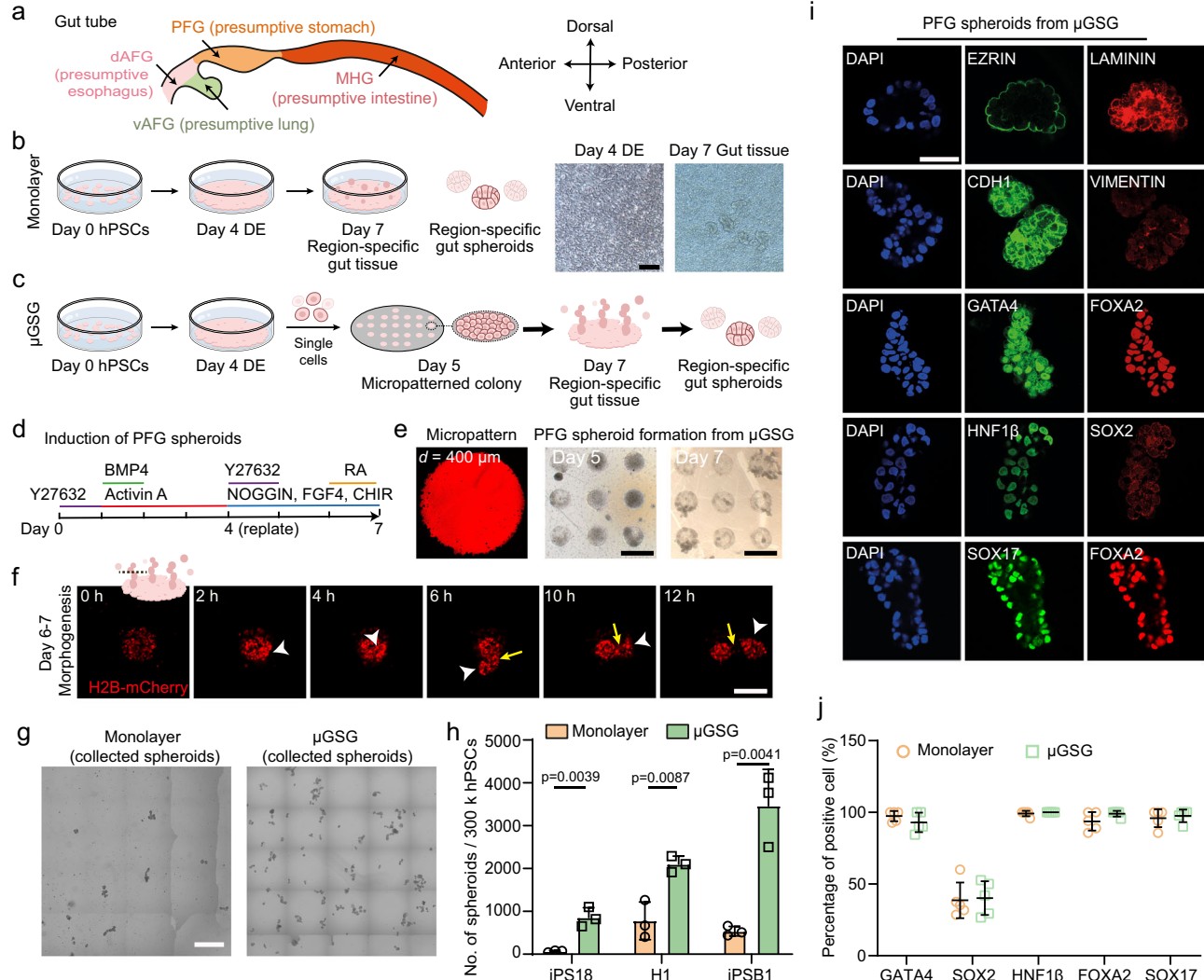

**Fig. 1 | Micropatterned gut spheroid generator (µGSG) enhances the biogenesis of posterior foregut (PFG) spheroids from human pluripotent stem cells (hPSCs).** **a** Schematic showing endoderm patterning and region-specific organo-genesis of the gut tube. PFG: posterior foregut. dAFG: dorsal anterior foregut. vAFG: ventral anterior foregut. MHG: mid-hind gut. **b** Schematic and representative phase contrast images of conventional monolayer-based induction of gut spheroids through an intermediate definitive endoderm (DE) stage. Scale bar: 100 µm. *n* = 3 independent experiments. **c** Schematic of the generation of gut spheroids in µGSG. **d** Induction of PFG spheroids. **e** Representative fluorescence image of micropattern (left) and phase contrast images of cell culture in µGSG on day 5 (middle) and day 7 (right). Scale bar: 500 µm. *n* = 3 independent experiments. **f** Representative time-lapse confocal micrographs showing PFG spheroid morphogenesis from days 6 to 7. (Inset) Schematic showing the optical section selected for viewing tissue fission. H1 H2B-mCherry cells and µGSG with a diameter of 200 µm were used. White

arrowheads indicate newly formed spheroid. Yellow arrows indicate the tissue fis-sion site. Scale bar: 200 µm. Similar results were seen in *n* = 3 independent experiments. **g** Representative phase contrast images showing PFG spheroids col-lected from monolayer-induction (left) and µGSG (right) on day 7, respectively. Scale bar: 500 µm. **h** Bar plot showing the number of PFG spheroids generated per 300,000 hPSCs input by monolayer-induction and µGSG, using three hPSC lines (iPS18, H1, and iPSB1), respectively. Data were plotted as mean ± s.d. *n* = 3 inde-pendent experiments. **i** Representative confocal micrographs showing immunos-taining of indicated markers in µGSG-derived PFG spheroids. DAPI (blue) stains the nuclei. Scale bar: 50 µm. **j** Scatter plot showing the percentages of cells positive of indicated markers in PFG spheroids derived from monolayer-induction and µGSG, respectively. Data were plotted as mean ± s.d. *n* = 5 independent experiments. *P*-values were calculated using unpaired, two-sided Student's *t*-test. Source data are provided as a Source Data file.

different from those of tissue patterning, folding, and lumenogen-esis seen in previous models[17–24], remains undetermined. Here we developed a micropatterned gut spheroid generator (µGSG) system for efficient generation of various region-specific gut spheroids from hPSCs. Specifically, in contrast to conventional monolayer-based induction without extrinsic mechanical guidance (Fig. 1b), we used micropatterns to implement controls on the size and shape of colonies of hPSC-derived definitive endoderm (DE) cells, which are source cells for generation of gut spheroids upon further induction and region-specific differentiation (Fig. 1c)[25–27]. Our findings show that µGSG surpasses the suboptimal efficiency of conventional monolayer-based induction, providing universal improvement of

the biogenesis of gut spheroids corresponding to various regions of the gut tube. Notably, we demonstrate that µGSG-enhanced bio-genesis of gut spheroids is independent of micropattern shape and size, but driven by mechanically enforced cell multilayering and crowding, suggesting a micropattern edge-insensitive, buckling-like mechanism to regulate initial tissue budding from originally flat gut tissues. We further combine experimental findings and an active-phase-field theory to recapitulate morpho-mechanics of subsequent spheroid pearling and fission. This work provides an efficient, universal, scalable, and standardized system to produce human gut spheroids, and unveils mechanobiology paradigms based on tissue architecture and surface tension for controlling

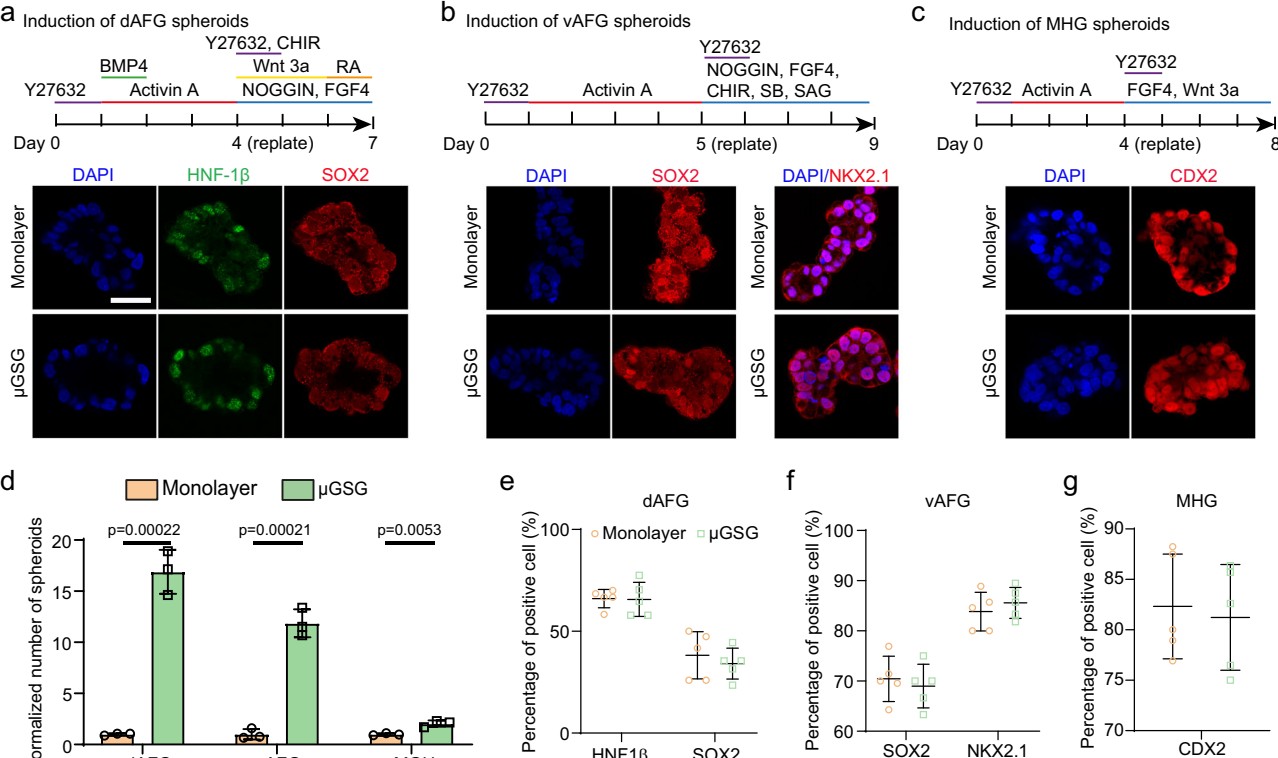

**Fig. 2 | Micropatterned gut spheroid generator (µGSG) enhances the biogenesis of different types of region-specific gut spheroids from human pluripotent stem cells. a–c** Induction of dAFG (dorsal anterior foregut) (**a**), vAFG (ventral anterior foregut) (**b**), and MHG (mid-hind gut) (**c**) spheroids (upper) and representative confocal micrographs (lower) showing the staining of indicated markers in spheroids derived from indicated conditions. Scale bar: 50 µm. *n* = 5 independent experiments. **d** Bar plot showing normalized number of dAFG, vAFG and MHG spheroids generated by monolayer-induction and µGSG, respectively, using iPS18 cell line. *n* = 3 independent experiments. **e–g** Scatter plots showing percentages of cells positive of HNF1β and SOX2 in dAFG (**e**), SOX2 and NKX2.1 in vAFG (**f**), and CDX2 in MHG (**g**) spheroids derived from indicated conditions. *n* = 5 independent experiments. Data were plotted as mean ± s.d. *P*-values were calculated using unpaired, two-sided Student's *t*-test. Source data are provided as a Source Data file.

tissue morphogenesis, thereby opening new avenues for advancing tissue engineering and regenerative medicine.

## Results

### µGSG universally enhances the biogenesis of region-specific gut spheroids

In the µGSG system, a human induced pluripotent stem cell (hiPSC) line, iPS18, was firstly differentiated to FOXA2 + /SOX17 + DE cells as previously reported[28], with ~100% efficiency (Supplementary Fig. 1a). The DE cells were then dissociated and plated as single cells at 350,000 cells cm$^{-2}$ onto circular adhesive micropatterns ($d$ = 400 µm unless noted otherwise) on day 4 (Fig. 1c). Upon further differentiation towards a posterior foregut (PFG) identity (presumptive stomach region; Fig. 1d), distinct tissue morphogenesis was evident in µGSG, featuring finger-like tissue columns emerging from flat micropatterned colonies on day 6, followed by spheroid formation through tissue fission on day 6-7 (Fig. 1e, f; Supplementary Fig. 2; Supplementary Video 1). Herein, we specifically defined an "input-output ratio" to measure the biomanufacturing efficiency of gut spheroid generation, which is calculated as the ratio between the total number of spheroids generated and the number of initial hPSC input on day 0 (see Methods for details). Notably, µGSG showed significantly enhanced generation of spheroids from iPS18 (~850 spheroids per 300,000 hPSC input), as compared to limited spheroid formation using monolayer-based induction (~60 spheroids per 300,000 hPSC input) (Fig. 1g, h). Similar improvement is also observed in µGSG-based biogenesis of PFG spheroids using a human embryonic stem cell (hESC) line, H1, and another hiPSC line, iPSB1, respectively (Fig. 1h; Supplementary Fig. 1b). µGSG-derived PFG

spheroids (µGSG-PFG) exhibit a polarized epithelial structure, with EZRIN+ apical surfaces facing outside and LAMININ+ basal regions embedded in the cell cluster (Fig. 1i; Supplementary Fig. 3), showing structural features comparable to those generated from monolayer-induction (Supplementary Fig. 4). µGSG-PFG also express GATA4, SOX2, HNF1β, FOXA2, and SOX17 at levels comparable with those derived using monolayer-induction, suggesting molecular fidelity of µGSG-PFG (Fig. 1i, j; Supplementary Figs. 3 and 4). Of note, treating monolayer-based culture with the same chemical treatment scheme and timing as in µGSG (in particular, adding Y27632 from day 4-5) did not show notable improvement in spheroid generation (Supplementary Fig. 5a). µGSG using substrates fabricated via microcontact printing or direct protein adsorption both showed enhanced spheroid generation, suggesting the robustness of this method on differently treated substrates (Supplementary Fig. 5b).

The µGSG can also be applied to generate spheroids corresponding to other gut tube regions such as dorsal anterior foregut (dAFG, with an esophageal fate), ventral anterior foregut (vAFG, with a lung fate), and mid-hind gut (MHG, with an intestinal fate), respectively (Fig. 2a, b). Compared with monolayer-induction, µGSG shows generally greater efficiencies in generating dAFG, vAFG, and MHG spheroids (Fig. 2d; Supplementary Fig. 6). These spheroids, henceforth termed µGSG-dAFG, µGSG-vAFG, and µGSG-MHG, respectively, also exhibit similar expression levels of region-specific markers as their counterparts derived from monolayers (Fig. 2e–g; Supplementary Fig. 7).

To examine the functional fidelity of gut spheroids derived from µGSG, we embedded µGSG-PFG in 3D extracellular matrix to drive their continuous gastric differentiation[25] (Fig. 3a, b). The µGSG-PFG

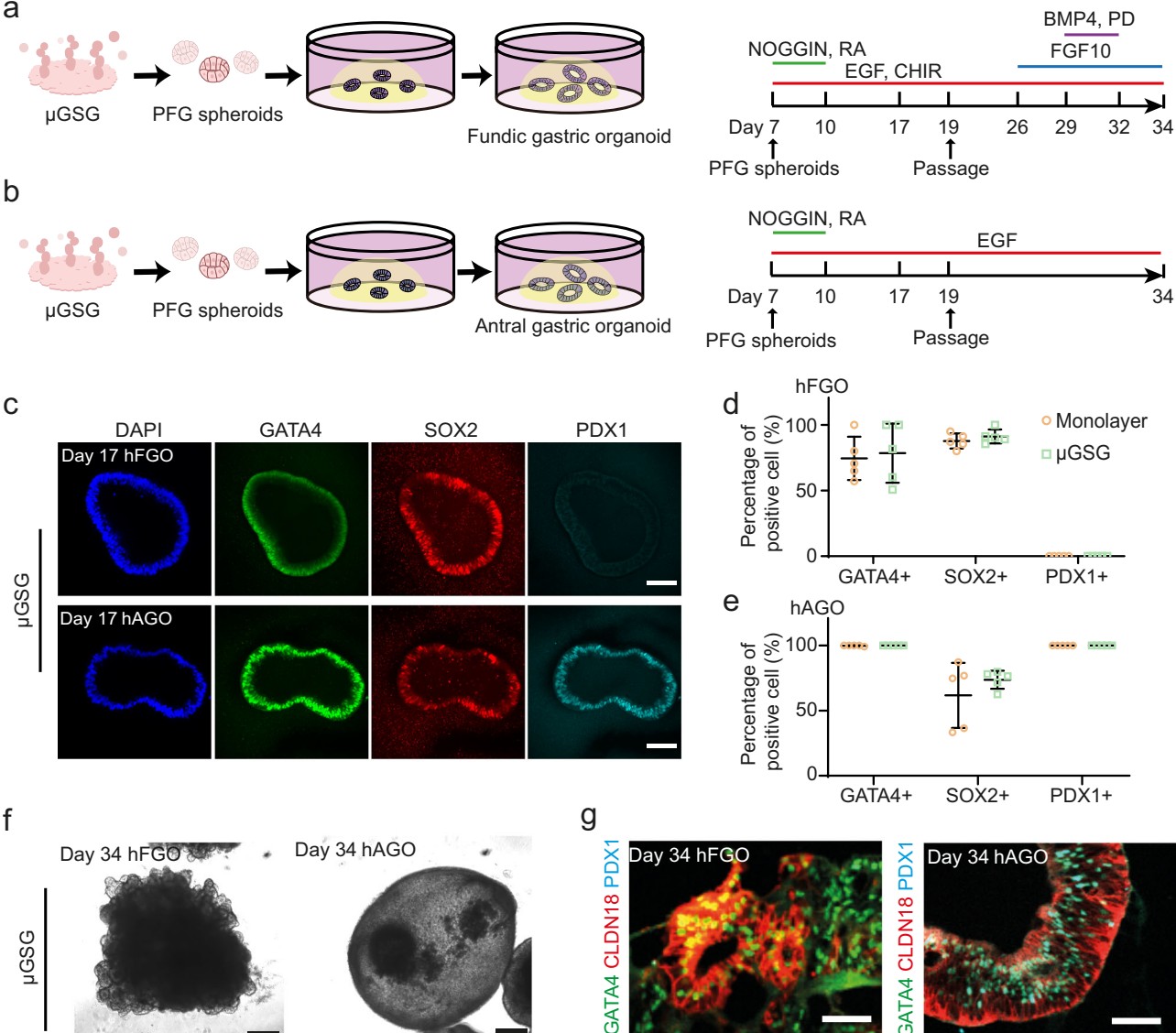

**Fig. 3 | Micropatterned gut spheroid generator (µGSG)-derived gut spheroids possess region-specific developmental potential. a**, **b** Differentiation of µGSG-derived PFG (posterior foregut) spheroids (µGSG-PFG) to fundic (**a**) and antral (**b**) gastric organoids, respectively. **c** Representative confocal micrographs showing staining of the nuclei (DAPI), GATA4, SOX2, and PDX1 in day 17 hFGO (human fundic gastric organoid, upper) and hAGO (human antral gastric organoid, lower) generated from µGSG-PFG. Scale bar: 100 µm. **d**, **e** Scatter plots showing percentages of cells positive of GATA4, SOX2, and PDX1 in day 17 fundic (**d**) and antral (**e**) organoids differentiated from monolayer-PFG and µGSG-PFG, respectively. Data were plotted as mean ± s.d. $n = 5$ independent experiments. **f** Representative phase contrast images of day 34 hFGO (left) and hAGO (right) differentiated from µGSG-PFG. Similar results were observed in $n = 3$ independent experiments. Scale bar: 200 µm. **g** Representative confocal micrographs showing immunostaining of GATA4, CLDN18, and PDX1 in day 34 hFGO (left) and hAGO (right) differentiated from µGSG-PFG. Similar results were observed in $n = 3$ independent experiments. Scale bar: 50 µm. Source data are provided as a Source Data file.

successfully differentiated to fundic (GATA4 + /SOX2 + /PDX1-) and antral (GATA4 + /SOX2 + /PDX1 + ) gastric organoids, respectively, upon domain-specific inductions[3,25,29] (Fig. 3c–e), similar to the differentiation of monolayer-induced spheroids (Supplementary Fig. 8). Extended differentiation of µGSG-derived organoids also induced gland-like budding morphogenesis in fundic, but not antral, gastric organoids (Fig. 3f), as well as ubiquitous expression of gastric-specific claudin (CLDN18) (Fig. 3g). Notably, the µGSG-dAFG and µGSG-vAFG are capable of generating SOX2 + /P63+ esophageal organoids[4] and SOX2 + /NKX2.1+ lung organoids[5], respectively (Supplementary Fig. 9a–d). The µGSG-MHG could also give rise to intestinal (CDX2 + ) and colonic (SATB2 + ) organoids[30], respectively (Supplementary Fig. 9e–h). These data show that µGSG-derived gut spheroids possess proper developmental potential for generating region-specific organoids.

As another strategy for generating multicellular spheroids with high efficiency, forced aggregation of singly dissociated cells has been widely adopted with the help of a number of engineering tools (e.g., microwells, hanging drops, etc.). To compare the performance of µGSG and that of forced cell aggregation in generating gut spheroids, we used custom-made U-bottom microwells to enforce the aggregation of singly dissociated PFG cells on day 7. Notably, forced cellular aggregation only produced spheroids that heterogeneously express PFG markers such as SOX17 and CDH1 (Supplementary Fig. 10a). In addition, extended culture of these spheroids till day 17 only yielded FGO with inconsistent expression of SOX2 and GATA4 (Supplementary Fig. 10b). In contrast, µGSG-derived PFG and FGO exhibit uniform expression of corresponding markers (Supplementary Fig. 10). This is probably due to the self-selective nature of spontaneous gut spheroid morphogenesis from an otherwise heterogeneous gut cell

population[31–34]. Therefore, these data suggest that forced cellular aggregation, despite its merits in efficiency, only produces gut spheroids that are suboptimal in biological fidelity and developmental potential compared to that derived from μGSG.

Together, the above findings demonstrate μGSG as a scalable, standardized, and generally applicable system that could enhance the generation of region-specific, high-fidelity gut spheroids from hPSCs.

### Micropattern geometry-insensitive, mechanically enforced cell multilayering and crowding is necessary and sufficient for enhancing gut spheroid biogenesis

To explore the mechanism underlying such mechanically enhanced biogenesis of gut spheroids in μGSG, we used μGSG-PFG as a model system henceforth. We first examined whether it is subject to the influence of micropattern size and geometry, which have been found important in guiding embryoid and organoid development[17–19,35–39]. Interestingly, spheroid formation efficiency remained unchanged in μGSG featuring circular micropatterns with a broad range of diameters ($d = 50–8000$ μm), all showing significantly higher efficiency than monolayer-based induction (Fig. 4a, b; Supplementary Fig. 11a). Spheroid formation efficiency also remains unaffected in μGSG made of rectangular micropatterns with different aspect ratios and areas (Fig. 4c, d). Therefore, μGSG enhances the biogenesis of gut spheroids independent of the size and shape of the micropatterns, implicating a previously unappreciated mechanism unlike the canonical edge-sensing mechanism reported in other micropatterned-based models[17,23,35,39–42].

The geometry-independence of gut spheroid formation in the μGSG suggests a possible mechanism for the enhanced biogenesis of gut spheroids through micropattern-induced spatial confinement, which might enforce cellular crowding and organization into multi-layers and thereby facilitate subsequent spheroid initiation. Indeed, significant cell multilayering and greater colony thickness were observed in μGSG on day 5; in contrast, thinner cell sheets and limited multilayering were seen in monolayer-based induction (Fig. 4e, f). Consistently, plating cells at 200,000 and 100,000 cm$^{-2}$ in μGSG led to notably less multilayering and decreased spheroid formation efficiency, in contrast to the multilayering and efficient spheroid formation under high DE cell seeding density conditions (e.g., 350,000 and 300,000 cells cm$^{-2}$), (Fig. 4g–i, Supplementary Fig. 12). Of note, different hPSC plating density in monolayer-based induction did not result in notable improvement of spheroid generation[16,27], suggesting a fundamental distinction between monolayer-induction and the μGSG, with the latter featuring a double cell-plating protocol. The multi-layering in μGSG also induced notable cellular crowding and intra-tissue compressive deformation, as reflected by the reduced size of cell nuclei under the higher plating density of DE cells at day 5 (Fig. 4j). These data suggest that mechanically enforced cell multilayering and crowding is necessary for enhanced spheroid formation by μGSG.

Furthermore, we observed a notable correlation between increased tissue thickness, as well as tissue crowding (i.e., internal compressive deformation reflected by the reduced size of nuclei), and enhanced initiation of tissue buds from originally flat gut tissues in μGSG from day 5-6 (Fig. 4k–m), which precedes the improved gut spheroid generation. Given these observations, we hypothesize that μGSG might leverage the synergy of increased tissue thickness and internal compressive deformation to enhance the efficiency of gut spheroid generation by facilitating the initiation of tissue buds from originally flat gut tissues via a buckling-like mechanism. To examine this hypothesis, we developed a biomechanical model to recapitulate the architecture of PFG tissues featuring a thin, stiff F-actin-rich cortex atop a thick, soft tissue layer based on our experimental observation (Fig. 5a; see Methods for details). The modulus of the F-actin-rich cortex and the underlying tissue bed were defined as $10^4$ Pa and $10^1$ Pa, respectively, based on previous measurements[43]. Upon increasing

internal compressive deformation, this model could undergo buckling-like morphogenesis that resembles gut tissue budding seen on days 5–6 (Fig. 5a; see Methods for details). Specifically, using the finite element method, we simulated tissue buckling behaviors driven by different levels of internal compressive strain in this model (Fig. 5b, c; see Methods for details). Our simulation results indicate that greater tissue thickness could facilitate the onset of tissue buckling under low-level tissue compression, and it further promotes tissue buckling amplitude under greater compressive strain. Together, these data substantiated that tissue multilayering and crowding might be essential mechanisms for enhancing gut spheroid formation via promoting tissue bud initiation through a buckling-like mechanism.

To examine whether such mechanically enforced multilayering and crowding is also sufficient to enhance the biogenesis of spheroids under culture conditions that otherwise show limited spheroid formation, we conducted a set of scratch-based assays using conventional monolayer-induction (Fig. 6a). Specifically, our scratch-reattach (SR) assay used a pipette tip to scratch and lift the DE cell sheet on day 4, leaving the lifted cell sheet to contract, curl back, and reattach to the intact cell monolayer next to the scratched area, which resulted in notable cell multilayering (Fig. 6a–c). In contrast, in the scratch-clear (SC) assay, we immediately cleared the lifted cell sheet after scratch, using an aspiration glass pipet under microscope, thereby leaving only cell monolayer next to the path of scratch, which also exhibit less tissue crowding compared to that in the SR assay (Fig. 6a–d). Inter-estingly, upon further differentiation, prominent spheroid formation was observed in the SR, but not in the SC or monolayer condition, with spheroid morphogenesis occurring mostly at the site of SR-induced cell multilayering (Fig. 6a, e). These findings clearly suggest that mechanically enforced cell multilayering and crowding is a generic, yet previously underappreciated, methodology both necessary and suffi-cient for enhancing the biogenesis of gut spheroids in various conditions.

### An active phase field model recapitulates the instability-driven morphomechanics of monodispersed gut spheroid fission

Notably, above experiments using μGSG with different sizes and shapes (Fig. 4a, c) also showed that gut spheroids exhibit conserved, mono-dispersed morphological features (e.g., width, aspect ratio, and projected area) (Fig. 7a–c; Supplementary Figs. 11b–d & 13). Unlike micropattern geometry-dependent morphogenic mechanisms pre-viously reported in other models[17–19,35–37,39], these findings suggest that the final, fission-like morphogenic step of gut spheroid in μGSG (Fig. 1f) is dictated by a tissue-intrinsic, yet unknown, mechanism. Given the dramatic tissue deformation during gut spheroid fission, we examined tissue-level forces and cytoskeletal remodeling during spheroid mor-phogenesis. Specifically, we conducted time-course analysis of the localization of actomyosin contractile machinery. It clearly indicates a gradual accumulation of both F-actin and phosphorylated myosin light chain (pMLC) at the outer surface of pre-fission tissue columns from day 5 to 7 (Fig. 7d; Supplementary Fig. 14a). Although F-actin accu-mulation was also seen at the lumenal surface of pre-fission tissue columns, simultaneous enrichment of pMLC was absent (Fig. 7d; Supplementary Fig. 14a). These data implicate that a gradual build-up of actomyosin machinery and tissue tension at the outer, but not inner, surface of tissue column might be a key driver for spheroid morpho-genesis in μGSG.

To test the above hypothesis, we developed a theoretical model from a reductionist viewpoint (Fig. 7e; see Methods for details). This model considers spheroid formation from an initially straight tissue column, and takes into account the two potential driving mechanisms implied by the above experiments: (1) the accumulation of compo-nents of actomyosin machinery at the outer tissue surface, depicted by an active migration coefficient $\lambda$ in this model; and (2) the resultant tissue surface tension due to myosin motor functions, denoted by

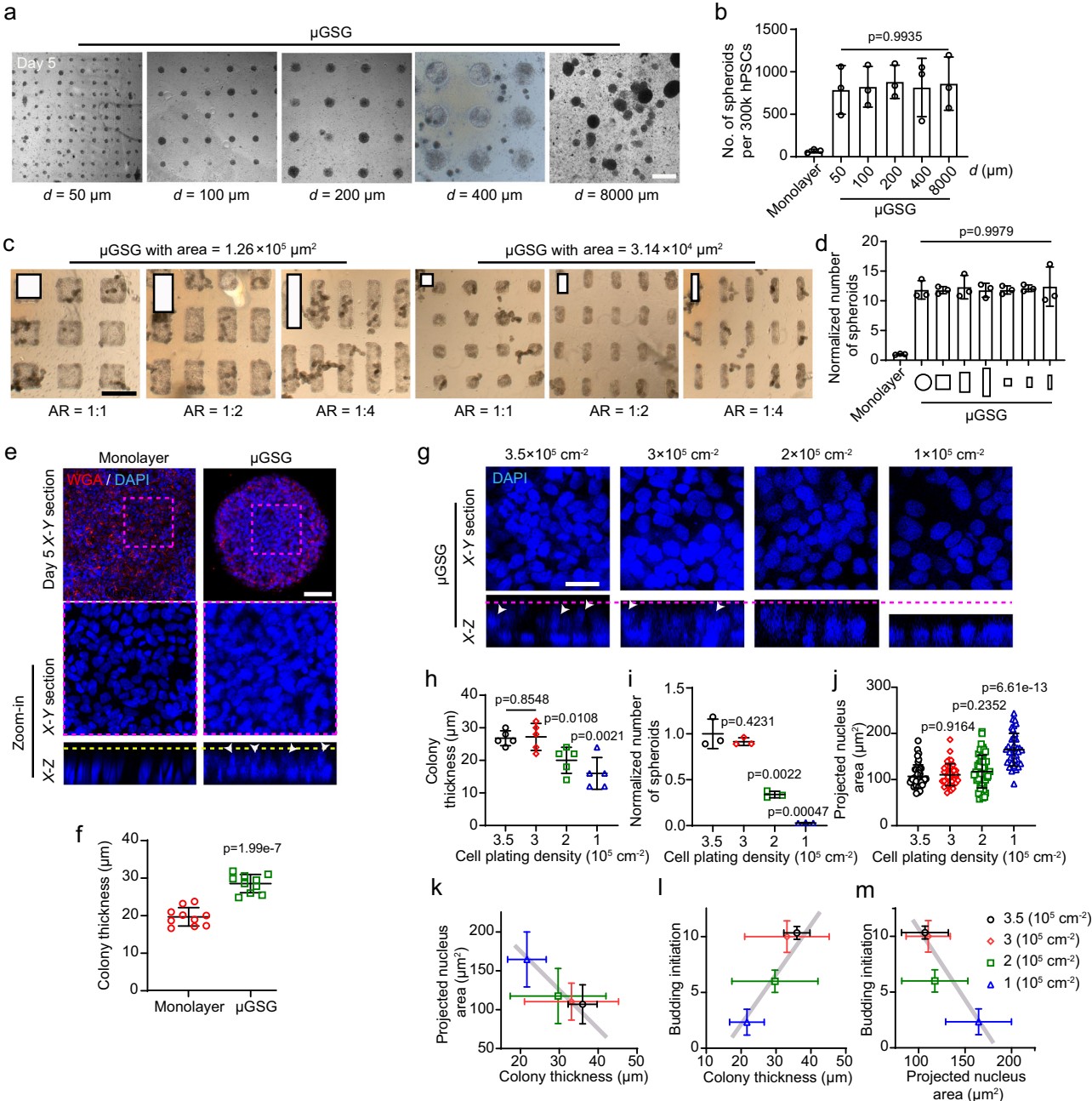

**Fig. 4 | Mechanically enforced cell multilayering and crowding is necessary for enhancing the biogenesis of gut spheroids. a** Phase images showing cell culture in μGSG (micropatterned gut spheroid generator) with different diameters, *d*, on day 5. Scale bar: 400 μm. **b** The number of PFG (posterior foregut) spheroids generated in indicated conditions. *n* = 3 independent experiments. **c** Phase images showing cell culture in μGSG with rectangular micropatterns of different aspect ratios (AR) and areas on day 7. Scale bar: 400 μm. **d** Normalized number of PFG spheroids generated from monolayer, circular μGSG (*d* = 400 μm) and rectangular μGSG shown in (**c**). *n* = 3 independent experiments. **e** Confocal micrographs showing the X-Y and X-Z sections of nuclei in indicated conditions on day 5. Wheatgerm agglutinin (WGA) stains cell membrane. Purple rectangles mark areas for zoom-in. White arrowheads indicate cell multilayering. Yellow dashed lines show the upper boundary of cell multilayering. Scale bar: 100 μm. **f** Colony thickness in indicated conditions on day 5. *n* = 5 independent experiments, each with two technical replicates. **g** Confocal micrographs showing the X-Y and X-Z sections of

nuclei in μGSG under indicated conditions on day 5. White arrowheads indicate cell multilayering. Purple dashed lines show the upper boundary of cell multilayering. Scale bar: 100 μm. **h** Scatter plot showing colony thickness in indicated conditions on day 5. *n* = 5 independent experiments. **i** Normalized number of spheroids generated by μGSG under indicated conditions. *n* = 3 independent experiments. **j** Projected nucleus area in indicated conditions. *n* = 4 independent experiments, each with at least ten randomly selected views. **k** Correlation between tissue thickness and projected nucleus area shown in (**j**). *n* = 5 independent experiments for colony thickness analysis. **l**, **m** Correlation between initiation of tissue buds and (**l**) tissue thickness (data from (**k**)), and (**m**) projected nucleus area (data from (**j**)). *n* = 3 independent experiments for budding initiation analysis. All data were plotted as mean ± s.d. *P*-values were calculated using one-way analysis of variance (ANOVA) and unpaired, two-sided Student's t-test. Source data are provided as a Source Data file.

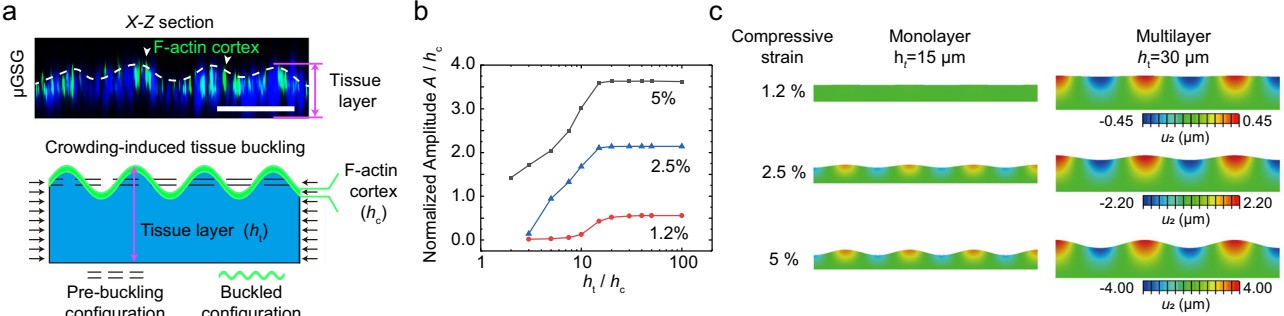

**Fig. 5 | Mechanically enforced cell multilayering and crowding are essential for enhancing the initiation of gut tissue budding via a buckling-like mechanism.** **a** (upper) Representative X-Z section showing the architecture of the gut tissue cultured in µGSG (micropatterned gut spheroid generator), with F-actin stained in green and nuclei in blue. The buckled configuration of F-actin cortex was outlined with the white dashed line. Similar results were seen in $n = 3$ independent experiments. (lower) Schematic of the tissue architecture shown in the upper panel, which was used to conduct finite element analysis of tissue buckling under crowding-induced compressive strain (see Methods for detailed descriptions). **b** Simulation results showing the change in normalized buckling amplitude as a function of the tissue-to-cortex thickness ratio ($h_t / h_c$). Results obtained under 1.2%, 2.5%, and 5% lateral compressive strain was shown, respectively. **c** Contours of tissue deformation under different levels of compressive strain for monolayer tissue ($h_t = 15$ µm) and multilayer tissue ($h_t = 30$ µm), respectively. Displacement field in the $y$ direction ($U_2$) was color-coded as shown by the color bars. Source data are provided as a Source Data file.

surface tension coefficient $\kappa$. For simplicity, we focused on a two-dimensional analysis of this model. However, due to the difficulty in using solid mechanics-based models to recapitulate such fluid-like behaviors of tissue fission, it requires a new strategy.

Herein, to quantitatively analyze how the above mechanisms might drive spheroid fission, we developed a theoretical framework based on the active phase field method[44] (see Methods for details). In this active-phase-field model, intrinsic activity refers to the spontaneous energy transformation from chemical energy to mechanical energy, which enables tissue movement and deformation. To simulate the fluid-like tissue deformation during fission, the system is described by a scalar field $\phi$, where $\phi = 1$ denotes the tissue center, $\phi = -1$ the culture medium, and $\phi = 0$ the outer surface of the tissue (Fig. 7e). The diffusion kinetics of $\phi$ is dictated by the chemical potential, $\mu$, of the system, as $\partial_t \phi = M\nabla^2 \mu$, wherein $M$ is the diffusion coefficient. The chemical potential is determined by $\mu = \frac{\delta F}{\delta \phi} + \lambda(\nabla\phi)^2$, for which the first term is the variational derivative of a surface tension-dependent term of free energy, $F = \int[-\frac{a}{2}\phi^2 + \frac{a}{4}\phi^4 + \frac{\kappa}{2}|\nabla\phi|^2]d^2r$. The first two terms of the free energy function $F$ are Landau free energy, which is a double-well potential function whose minima are divided at $\phi = \pm 1$ to distinguish the two phases (i.e., tissue vs. medium). The parameter $a$ describes the magnitude of the Landau free energy, with greater $a$ to reflect a greater energy barrier between the two phases. The third term of $F$ is the surface energy to maintain the minimum surface area in the case of volume conservation, which reflects the active contraction behavior at the tissue surface. Parameter $\kappa$ is the surface tension coefficient that describes the magnitude of the surface energy and measures the strength of the contraction ability at the interface, which has been shown to be important in morphogenesis[45,46]. The second term of the chemical potential $\mu$ describes the active migration of the scalar field $\phi$ (i.e., the change in the spatial distribution of cytoskeletal components for actomyosin contractility; reflected by active migration coefficient $\lambda$). In general, we assume that the active migration coefficient $\lambda < 0$, which leads to a lower chemical potential at the surface and $\phi$ will migrate toward the surface, representing the directional accumulation of actomyosin machinery (e.g., F-actin and pMLC) at the outer tissue surface. By solving the time-dependent evolution of the scaler field $\phi$, we could depict the dynamics of tissue morphogenesis by tracing the tissue boundary at $\phi = 0$ (see Methods for details).

Notably, simulation results from our model clearly demonstrate that spheroids form through an instability-driven process (Fig. 7f). Specifically, originally flat tissue surface first undergoes buckling to form undulating surfaces, resembling the intermediate morphogenic

process experimentally observed in pre-fission tissue column. Post-buckling morphodynamics features further tissue fission occurring at distinct "necking" regions defined by previous buckling, resulting in the separation of tissue parts that eventually form individual spheroids (Supplementary Video 2). Our model also successfully recapitulates the conserved, monodispersed spheroid fission, which shows limited variation in spheroid size as seen in experiments (Fig. 7a–c & g; Supplementary Figs 11b, 13a). Noting the apparent resemblance, yet mechanistic distinctions, between the instability-driven fission of gut spheroids and the Rayleigh instability of passive fluids[47], it supports the notion to theorize tissue morphogenesis through analogy to the mechanics of active fluids[46,48].

## Biomechanical phase diagram of gut spheroid morphogenesis

To comprehensively understand how mechanical forces might regulate gut spheroid fission, we further explored the parametric space of our active phase field model. Based on simulation results obtained from a range of $\lambda$ and $\kappa$, we plotted a phase diagram of spheroid morphogenesis (Fig. 8a). In this diagram, smaller magnitudes of $\lambda$ and $\kappa$ denote lower levels of actomyosin accumulation and tissue tension at outer tissue surface. This phase diagram features three morphogenic states: (1) smooth state, where no tissue deformation occurs; (2) pearling state, where the tissue column forms undulations but cannot complete the fission; (3) fission state, where the necking regions in post-buckling tissue column are completely severed to form individual spheroids. Of note, this phase diagram clearly shows a mechanosensitive transition, depending on the values of $\lambda$ and $\kappa$, between "tissue pearling" and "tissue fission", suggesting that tissue surface tension might play a critical role in dictating the morphogenic states of gut tissues in µGSG.

To validate this theoretical prediction, we used small molecule inhibitor, Y27632, to disrupt the accumulation of actomyosin machinery and surface tension at outer tissue surface during gut spheroid fission. Indeed, treatment with 3 µM Y27632 was found to successfully inhibit the accumulation of both F-actin and pMLC at the outer surface of pre-fission tissue columns (Fig. 8b, c; Supplementary Fig. 14b). In the meantime, significant tissue "pearling" appeared in the µGSG under low-dosage treatment of Y27632, while tissue fission and spheroid formation were largely suppressed at the same dose (Fig. 8d, e; Supplementary Fig. 15), resembling the "fission to pearling" morphogenic state transition predicted by our theoretical model under reduced levels of surface tension and actomyosin accumulation. Furthermore, treatment with Blebbistatin, a Myosin IIA inhibitor, also

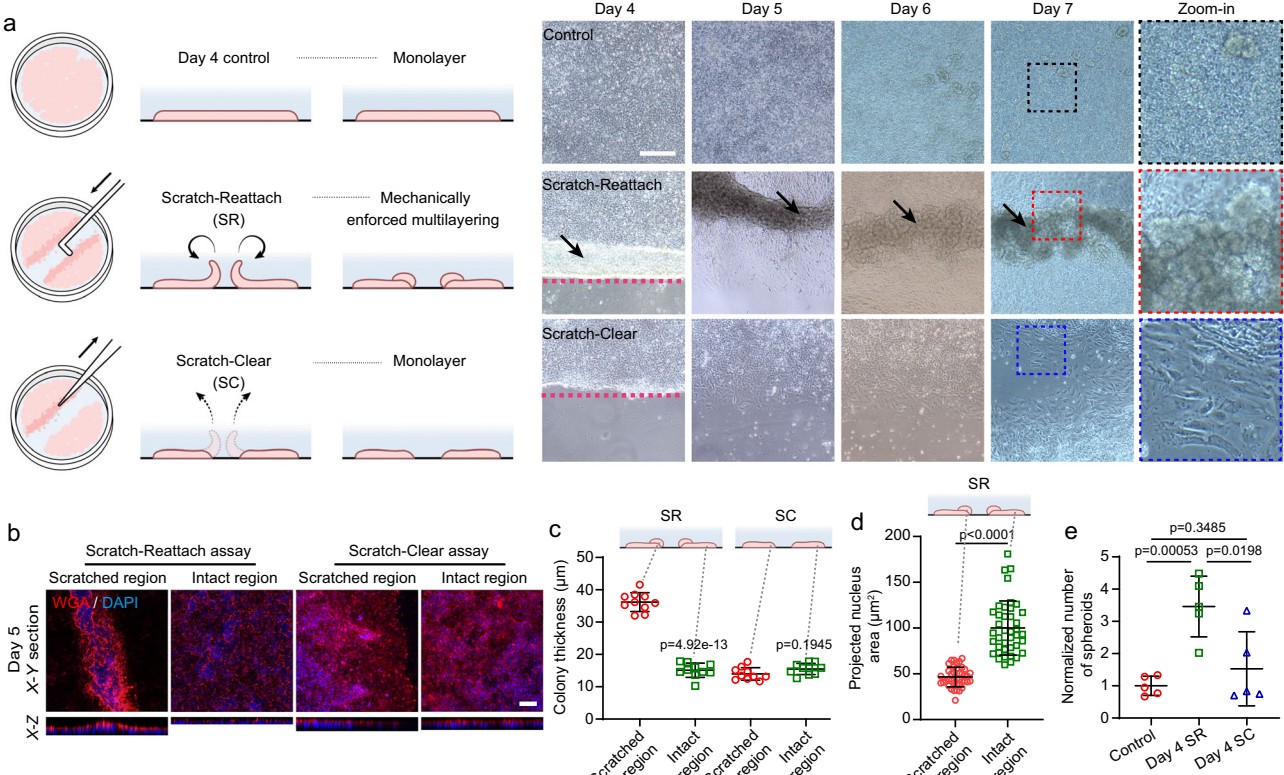

**Fig. 6 | Mechanically enforced cell multilayering and crowding is sufficient for enhancing the biogenesis of gut spheroids. a** Schematic showing inductions of posterior foregut spheroids in monolayer (upper), scratch-reattach (SR, middle), and scratch-clear (SC, lower) assays. Representative phase contrast images of cell culture under indicated conditions from day 4 to 7 were also shown. Red dashed lines show boundary of the scratched area. Black arrows indicate cell multilayering. Black, red, and blue rectangles mark areas shown in zoom-in images. Scale bar: 200 µm. Similar results were seen in $n = 5$ independent experiments. **b** Representative confocal micrographs showing the X-Y and X-Z sections of scratched and intact regions of day 5 cell culture in SR and SC assays, respectively. DAPI stains the nuclei

and wheat-germ agglutinin (WGA) stains the cell membrane. Scale bar: 100 µm. **c** Scatter plot showing colony thickness at the scratched and intact regions in SR and SC assays, respectively. $n = 5$ independent experiments, each with two technical replicates. **d** Scatter plot showing the projected area of nuclei at the scratched and intact regions in SR assay, respectively. $n = 4$ independent experiments, each with ten views randomly selected for analyzing the projected nucleus area. **e** Scatter plot showing the normalized number of spheroids generated from indicated conditions. $n = 5$ independent experiments. All data were plotted as mean ± s.d. *P*-values were calculated using an unpaired, two-sided Student's t-test. Source data are provided as a Source Data file.

resulted in similar fission to pearling transition, resulting in significant remnant tissue pearling and incomplete gut spheroid fission (Fig. 8d, e). As a result, both inhibition of actomyosin contractility and F-actin assembly showed a dose-dependent suppression of spheroid formation in µGSG (Fig. 8f–i). Together, the above theoretical and experimental results reveal a biomechanical phase diagram for gut spheroid fission and indicate the existence of a biomechanical "threshold" required for gut spheroid fission. These findings also substantiate the efficacy of our active phase field model for recapitulating important morphogenic states and underlying mechanisms during tissue morphogenesis.

## Discussion

In this work, we report µGSG as an efficient, scalable, and standardized system that is generally applicable for the generation of various region-specific gut spheroids – precursors for gastrointestinal and pulmonary organoids – from hPSCs, thereby revising and surpassing the sub-optimal efficiency of the monolayer-based method used in past 10 years for generating gut spheroids from hPSCs. Since first reported by Spence et al.[6], the monolayer-based induction of gut spheroids has been widely adopted for generating gastrointestinal and pulmonary organoids from hPSCs. However, the monolayer-based induction is limited by issues of reproducibility, efficiency, and standardization, which hinder the scalable production of organoids and barrier their broad applications in fundamental studies and translations[16]. On the

other hand, although forced aggregation of singly dissociated cells provides a meritorious method for efficient spheroid generation, it results in gut spheroids that lack sufficient biological fidelity and developmental potential compared to those generated via spontaneous spheroid morphogenesis. Our method – the µGSG – shows significantly improved efficiency in producing high-fidelity gut spheroids corresponding to four different gut tube regions, even using hPSC cell line that otherwise yields limited spheroids with monolayer-based induction. Therefore, the µGSG presents opportunities to facilitate translations towards scalable biomanufacturing of region-specific gut spheroids and gastrointestinal and pulmonary organoids in the future.

In recent years, there has been intense interest and rapid progress in applying micropatterns to control tissue morphogenesis in hPSC-based developmental models[17–19,35–39]. In these models, micropattern shape and size have been demonstrated important in guiding tissue morphogenesis via various kinds of edge-sensing mechanisms[17,23,35,39–42]. Challenging this dogma, our data show that the enhanced gut spheroid morphogenesis in the µGSG manifests a geometry-independent and edge-insensitive paradigm. Specifically, our study shows micropattern-enabled, mechanically enforced cell multilayering and crowding as a key mechanism both necessary and sufficient to enhance gut spheroid morphogenesis, via promoting tissue bud initiation through a buckling-like mechanism. This mechanism can also be extended to enhance spheroid morphogenesis in conditions that otherwise produce limited spheroids. This finding

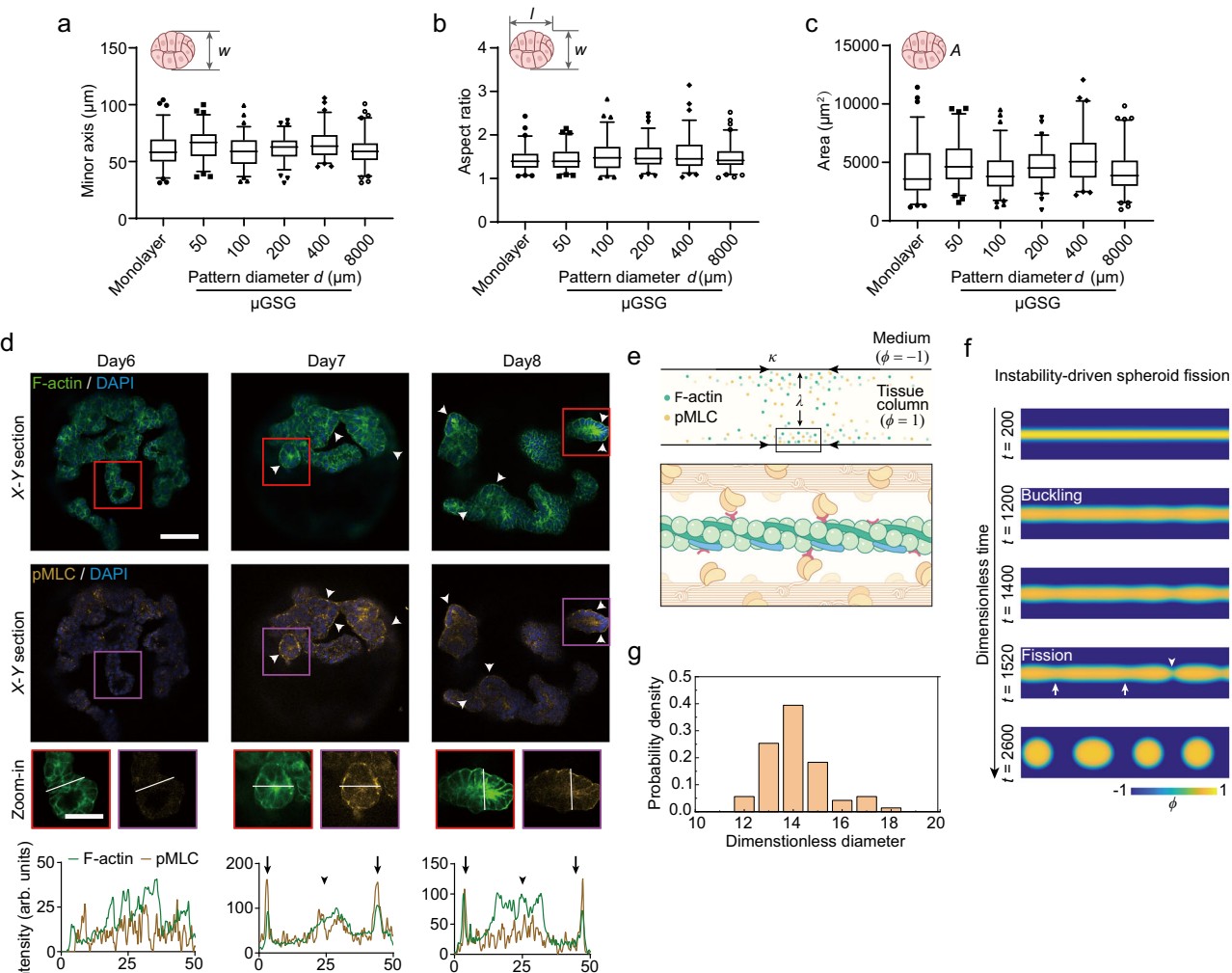

**Fig. 7 | Mono-dispersed gut spheroid fission is based on instability-driven morphomechanics. a–c** Box charts showing the minor axis length, w (**a**), aspect ratio, l/w (**b**), and projected area, A (**c**) of posterior foregut spheroids generated from monolayer and circular μGSG (micropatterned gut spheroid generator) with different diameter, d, (box: 25% − 75%, bar-in-box: median, and whiskers: 5% and 95%). $n_{spheroids} = 70$–82 for each independent experiment. $n = 3$ independent experiments. **d** (upper) Representative confocal micrographs showing X-Y sections of staining of F-actin, phosphorylated myosin light chain (pMLC), and the nuclei (DAPI) in protruding, pre-fission tissue columns formed in μGSG on indicated days with corresponding conditions. White arrowheads indicate the accumulation of F-actin and pMLC at the outer tissue surface. (middle) Representative tissue buds (prospective spheroid-forming regions; marked by red and magenta rectangles) were selected from pre-fission tissue columns and shown in zoom-in images.

(lower) Fluorescence intensities of F-actin and pMLC along the diameter of selected tissue buds (white lines in zoom-in images). Black arrows indicate locations of outer tissue surface, and black arrowheads show the location of inner / lumenal tissue surface, of selected tissue buds. $n_{pre\text{-}fission\ tissue\ columns} > 30$ and $n_{micropattern} > 5$ for each experiment. $n = 3$ independent experiments. Scale bar: 100 μm. **e** Schematic of the active-phase-field model of spheroid morphogenesis from a tissue column, featuring directional accumulation of F-actin and pMLC and the resultant tension at outer tissue surfaces (see main text and Methods for detailed descriptions). **f** Simulation results of spheroid fission, showing the onset of tissue buckling and subsequent tissue fission. White arrowhead indicates a representative fission site. White arrows show buckling-induced necking regions where fission will follow. **g** Probability density distribution of spheroid sizes predicted by the active-phase-field model. Source data are provided as a Source Data file.

not only lends an unconventional strategy for biomechanical control of organoid development and tissue morphogenesis, but also provides a unique view regarding the regulatory mechanism underlying mechanically guided cell organization.

Tissue fission is not only the final step in gut spheroid morphogenesis in vitro, but also important for organogenesis in vivo[49–51]. The mechanism underlying tissue fission has been believed to be based on programmed cell death[52,53]. However, experimental observations by us and others[49,54] challenge this classical view. Our findings show a previously unappreciated morphomechanics of tissue fission, wherein gradual build-up of both actomyosin machinery and tissue tension at tissue surface is key for spheroid pearling and fission through instability-driven process. Combining experiments and theoretical modeling, our data demonstrate instability-driven, monodispersed tissue fission, as well as a biomechanical phase diagram governing such

morphomechanical instability and force-dependent transition from "tissue pearling" to "tissue fission" during gut tissue formation, reflecting a unique biomechanical nature of tissue fission.

Epithelial tissue can drastically transform its shape during growth and development, exhibiting morphogenic behaviors more like fluids instead of solids. However, current theories for epithelial morphogenesis are mostly based on solid mechanics. Although attempts have been made recently to apply fluid mechanics to theorize epithelial morphogenesis[46], the consideration of the nature of epithelial tissue as "active" fluids, wherein the cytoskeletal remodeling and tissue tension work hand-in-hand to dictate the transformation of tissue morphology, remains limited and thus difficult to capture the morphodynamic evolution of tissue fission seen in experiments. Our study developed an active phase field model to recapitulate epithelial tissue morphogenesis. By considering the active transport of actomyosin machinery and

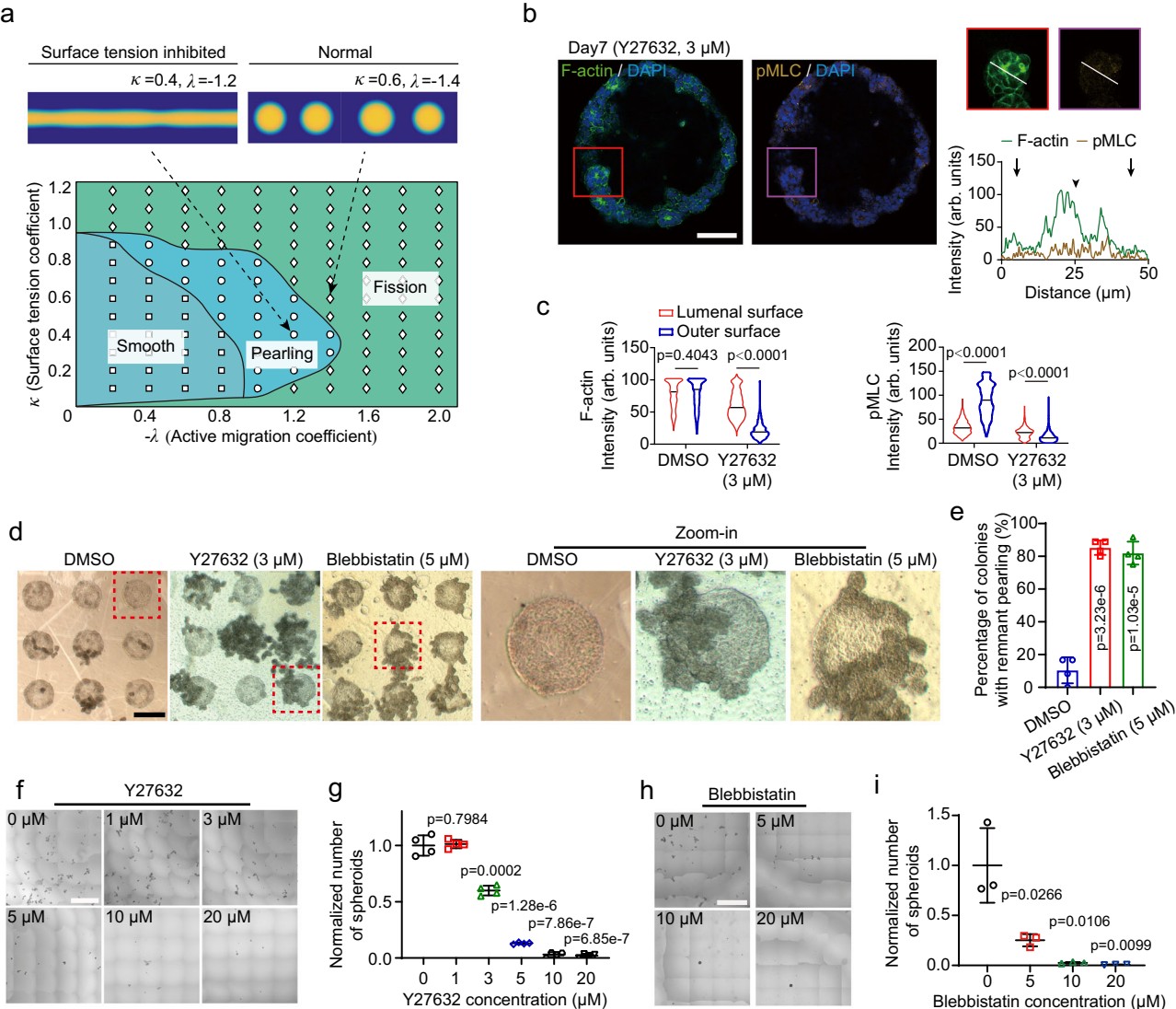

**Fig. 8 | Biomechanical phase diagram of gut spheroid fission. a** Phase diagram of different spheroid morphogenic states. (Inset) Simulation results are representative of spheroid morphogenesis with impaired surface tension (surface tension inhibited), or normal tension (normal), respectively. **b** (upper) Representative confocal micrographs showing X-Y sections of staining of F-actin, phosphorylated myosin light chain (pMLC), and the nuclei (DAPI) in protruding, pre-fission tissue columns formed in micropatterned gut spheroid generator with Y27632 treatment. (upper right) Representative tissue buds (prospective spheroid-forming regions) were selected from pre-fission tissue columns and shown in zoom-in images. (lower right) Fluorescence intensities of F-actin and pMLC along the diameter of selected tissue buds (white lines in zoom-in images). Black arrows indicate locations of outer tissue surface, and black arrowheads show the location of inner / lumenal tissue surface, of selected tissue buds. $n = 3$ independent experiments. Scale bar: 100 μm. **c** Violin plots showing average fluorescence intensities of F-actin and pMLC at the lumenal and outer surfaces of pre-fission tissue columns under indicated conditions on day 7. The lines in

the violin plots represent median values. $n_{\text{pre-fission tissue columns}} > 30$ and $n_{\text{micropattern}} > 5$ for each experiment. $n = 3$ independent experiments. **d** (left) Representative phase images showing posterior foregut tissues that remained attached to micropatterned gut spheroid generator after removing all readily-formed spheroids from the supernatant under indicated conditions on day 7. Scale bar: 400 μm. (right) Red dashed squares mark colonies for zoom-in. **e** Percentage of colonies with remnant pearling under the indicated conditions. $n = 4$ independent experiments. **f** Representative phase images showing readily-formed spheroids collected from μGSG under indicated conditions. Scale bar: 500 μm. **g** Normalized number of spheroids generated from indicated conditions. Data were plotted as mean ± s.d. $n = 4$ independent experiments. **h** Representative phase images showing readily-formed spheroids collected from μGSG under indicated conditions. Scale bar: 500 μm. **i** Normalized the number of spheroids generated from indicated conditions. Data were plotted as mean ± s.d. $n = 3$ independent experiments. *P*-values were calculated using an unpaired, two-sided Student's *t*-test. Source data are provided as a Source Data file.

the resultant tissue surface tension, our theoretical simulation shows the entire spectrum of gut spheroid morphodynamics as seen in experiments, and predicts its force-dependent state transitions. This theoretical framework might provide a conceptual and methodological foundation for studying tissue morphodynamics as active fluids.

In conclusion, we report herein a general platform, μGSG, that addresses the issues of efficiency, reproducibility, and standardization faced by conventional monolayer-based methods for generating gut spheroids from hPSCs. We demonstrate mechanically enforced cell multilayering and crowding as a general mechanism that is necessary

and sufficient to enhance gut spheroid generation in various conditions by facilitating tissue bud initiation via buckling-like process. We also show that the conserved, monodispersed tissue fission of gut spheroids is driven by an instability-based mechanism, which relies on the accumulation of actomyosin machinery and tension at tissue surfaces to dictate mechanosensitive transitions between different morphogenic states. In addition to advancing the scalable biomanufacturing of gut spheroids and organoids, this work also reveals previously underexplored morphogenic paradigms based on tissue architecture and surface tension for controlling tissue

morphogenesis and the biomechanical mechanisms underneath. Our active-phase-field morphomechanical model might also provide a versatile theoretical framework for studying tissue morphogenesis in a broad range of scenarios. Together, our results on both experimental and theoretical fronts might open new avenues to decipher and leverage the mechanobiological control over organoid development and tissue morphogenesis for fundamental research and translations.

## Methods

### Cell culture

Human pluripotent stem cell (hPSC) line used in this study included iPS18 (a human induced pluripotent stem cell, hiPSC, line, as a gift from Yang Zhao at the Peking University), H1 (a human embryonic stem cell, hESC, line; WA01; NIH registration number: 0043), and iPSB1 (an iPSC line, as a gift from Yang Zhao at the Peking University). hiPSCs and hESCs were cultured using feeder-free system and maintained on Geltrex™ (Thermo Fisher Scientific) coated plates in mTeSR™1 medium (STEMCELL Technologies). All hPSC lines were authenticated by original sources as well as in-house by immunostaining for pluripotency markers and successful differentiation to the three germ layers. The in-house authentication process was performed monthly. All hPSC lines were also verified as karyotypically normal and were tested monthly to ensure negative for mycoplasma contamination (LookOut Mycoplasma PCR Detection Kit, Sigma-Aldrich).

Human HEK293T cell lines were cultured in DMEM medium (Gibco, Catalog # 11995065) supplemented with 10% FBS (ExCell, Catalog # FND500) and 100 U of Penicillin-Streptomycin in 5% $CO_2$ at 37 °C.

### Fabrication of μGSG

μGSG was fabricated using microcontact printing technique as previously reported[55]. In brief, silicon templates of micropatterns with desired shapes and sizes were first fabricated using photolithography. To generate μGSG, cover glasses were pre-coated with a thin layer polydimethylsiloxane (PDMS; Sylgard 184, Dow-Corning) at base: curing agents ratio of 10: 1 (w / w). PDMS stamps with a diameter of 8 mm, which were generated from silicon templates through soft lithography, were incubated in 1% Geltrex™ diluted in DMEM/F12 for at least 1 h at 37 °C before blown dry using nitrogen gun. PDMS-coated cover glasses were activated by UV ozone for 7 min before being brought into conformal contact with Geltrex-coated PDMS stamps for at least 15 s[55]. After removing the PDMS stamp, PDMS-coated cover glasses were washed with DI water and then incubated with 0.5% Pluronics F127 solution (diluted in PBS) for 30 min, before being washed with PBS and stored at 4 °C until used.

### Fabrication of U-bottom microwells

The U-bottom agarose microwells were generated through a customer-made aluminum mold, which was designed with Autodesk Inventor Professional software and carved by a computer numerically controlled ultra-high precision lathe, for the fabrication of micropillars with hemispherical top. Both the diameter and height of micropillars were 400 μm, respectively. A 2% solution of ultrapure agarose (w/v, distilled water) was poured onto the aluminum mold, and then gently peeled off from the mold after solidification. The U-bottom agarose microwells were stored in PBS and sterilized with UV light for at least 30 min before being used for cell culture.

### Derivation of definitive endoderm

To derive definitive endoderm (DE) for the subsequent induction of PFG (posterior foregut) and dAFG (dorsal anterior foregut) differentiation, a 6-well plate of hPSCs that were around 80% confluent were digested as single cells using Accutase at 37 °C for 10 min, and then seeded onto a 24-well culture plate pre-coated with 1% Geltrex™ at 300,000 cells per well in mTeSR™1 medium containing 10 μM Y27632.

Afterwards, DE differentiation was initiated using day 1 differentiation medium (RPMI 1640 medium + 1% NEAA (non-essential amino acids) + 100 ng mL$^{-1}$ Activin A + 50 ng mL$^{-1}$ BMP4) for 24 h, followed by day 2 differentiation medium (RPMI 1640 medium + 1% NEAA + 0.2% defined fetal bovine serum (dFBS; Hyclone) + 100 ng mL$^{-1}$ Activin A) for another 24 h, then day 3 differentiation medium (RPMI 1640 medium + 1% NEAA + 2% dFBS + 100 ng mL$^{-1}$ Activin A) for another 24 h[25].

To derive DE for subsequent induction of vAFG (ventral anterior foregut) and MHG (mid-hind gut) differentiation, a 6-well culture plate of hPSCs that were around 85% confluent were dissociated, before being plated at 300,000–350,000 cells per well in mTeSR™1 medium containing 10 μM Y27632 in a 24-well culture plate coated with 1% Geltrex™. Afterwards, DE differentiation was initiated by using day 1 differentiation medium (RPMI 1640 medium + 1% NEAA + 100 ng mL$^{-1}$ Activin A) for 24 h, followed by day 2 differentiation medium (RPMI 1640 medium + 1% NEAA + 0.2% dFBS+100 ng mL$^{-1}$ Activin A) for another 24 h, then day 3 differentiation medium (RPMI 1640 medium + 1% NEAA + 2% dFBS + 100 ng mL$^{-1}$ Activin A) for another 24 h. DE differentiation was extended for another 24 h with day 4 medium (RPMI 1640 medium + 1% NEAA + 2% dFBS + 100 ng mL$^{-1}$ Activin A) in the case of vAFG induction[26,27].

### Gut spheroid generation by monolayer-based induction, μGSG, and forced cellular aggregation

During monolayer-based induction, DE cells were directly treated with a region-specific differentiation medium as specified below. For dAFG spheroid generation, cells were cultured in RPMI 1640 medium supplemented with 1% NEAA, 2% dFBS, 500 ng mL$^{-1}$ FGF4, 200 ng mL$^{-1}$ NOGGIN from day 4 to 7, together with 2 μM CHIR99021 from day 4 to 5, 500 ng mL$^{-1}$ Wnt 3a from day 4 to 6, and 2 μM RA (retinoic acid) from day 6 to 7[4]. For vAFG spheroid generation, cells were cultured in RPMI 1640 medium supplemented with 1% NEAA, 2% dFBS, 500 ng mL$^{-1}$ FGF4, 200 ng mL$^{-1}$ NOGGIN, 2 μM CHIR99021, 10 μM SB431542, and 1 μM SAG from day 5 to 9[27]. For PFG spheroid generation, cells were cultured in RPMI 1640 medium supplemented with 1% NEAA, 2% dFBS, 500 ng mL$^{-1}$ FGF4, 200 ng mL$^{-1}$ NOGGIN, and 2 μM CHIR99021 from day 4 to 7, with another 2 μm RA added from day 6 to 7[25]. For MHG spheroid generation, cells were cultured in RPMI 1640 medium supplemented with 1% NEAA, 2 mM L-glutamine, 2% dFBS, 500 ng mL$^{-1}$ FGF4, 500 ng mL$^{-1}$ Wnt 3a for 4 days[26]. The medium was replenished daily.

To generate gut spheroids using μGSG, DE cells were dissociated into single cells using Accutase at 37 °C for 5 min, before being seeded onto μGSG at 350,000 cells cm$^{-2}$ and then cultured in region-specific differentiation medium as those specified above. A total of 5–10 μM Y27632 was added for the first 24 h after replating DE cells on μGSG.

To generate gut spheroids using forced cellular aggregation, monolayer-based PGF tissue on day 6 was dissociated into single cells using Accutase at 37 °C for 5 min, before being seeded onto customer-made U-bottom agarose microwells at 100,000 cells per well and then cultured in PFG differentiation medium for day 6 to 7 containing 10 μM Y27632, before being used for further differentiation or molecular analyses.

### Quantification of spheroid formation

At the end-point of spheroid induction, readily-formed spheroids were all floating in the supernatant of the cell culture and were collected into centrifuge tubes. Both monolayer-based culture and μGSG-based culture were gently washed with DMEM/F12 three times to ensure that all floating spheroids are collected. Then we dispersed collected spheroids in a well of 24-well plate, before taking images of the whole well with Nikon Ti-E microscopy using bright field and mosaic imaging function. The number of collected spheroids in the whole well was counted using ImageJ 1.47. The efficiency of gut spheroid formation

was calculated as the ratio between the total number of spheroids and the number of initial hPSC input on day 0, i.e., input-output ratio. When noted, the normalized number of spheroids was calculated as the ratio of the total number of spheroids under certain conditions over the average number of spheroids under control conditions.

## 3D organoid culture

PFG spheroids were further differentiated to gastric organoids using protocols reported previously[25]. Briefly, spheroids were collected, resuspended in Geltrex™, and plated as 3D gel droplets with a volume of 20 μL. After solidification of Geltrex™ for 15 min at 37 °C, the 3D culture was initiated in basal gut medium, which contains: Advanced DMEM/F12, 1X N2 supplements (Invitrogen), 1X B27 supplements (without vitamin A, Invitrogen), 2 mM L-glutamine, 15 mM HEPES, and 100 U penicillin/streptomycin in combination with growth factors and/ or small molecules as specified below. For fundic gastric organoid differentiation, spheroids were first cultured in basal gut medium supplemented with 100 ng mL$^{-1}$ EGF, 200 ng mL$^{-1}$ NOGGIN, 2 μM RA, and 2 μM CHIR99021 for 3 days, then in basal gut medium supplemented with 100 ng mL$^{-1}$ EGF and 2 μM CHIR99021 until day 29, then in basal gut medium supplemented with 10 ng mL$^{-1}$ EGF and 2 μM CHIR99021 until day 34. From day 26 to 34, another 50 ng mL$^{-1}$ FGF10 was added to the differentiation medium. Another 50 ng mL$^{-1}$ BMP4 and 2 μM PD0325901 were pulsed from day 29 to day 32 in the differentiation medium[25]. Medium was replaced every 3-4 days. On day 19, organoids were passaged. For antral gastric organoid differentiation, the spheroids were cultured in basal gut medium supplemented with 100 ng mL$^{-1}$ EGF, 2 μM RA, and 200 ng mL$^{-1}$ NOGGIN for the first 3 days of 3D culture[25], then in basal gut medium supplemented with 100 ng mL$^{-1}$ EGF until day 29, then in basal gut medium supplemented with 10 ng mL$^{-1}$ EGF until day 34. Medium was replaced every 3–4 days. On day 19, organoids were passaged.

To generate esophageal organoids, dAFG spheroids were cultured in basal gut medium supplemented with 100 ng mL$^{-1}$ EGF, 200 ng mL$^{-1}$ NOGGIN, and 50 ng mL$^{-1}$ FGF10 for the first 3 days, then in basal gut medium supplemented with 100 ng mL$^{-1}$ EGF and 50 ng mL$^{-1}$ FGF10 for another 4 days. Then, organoids were cultured in basal gut medium containing 100 ng mL$^{-1}$ EGF until day 17[4]. Medium was replaced every 3–4 days.

To generate lung organoids, vAFG spheroids were cultured in 3D in basal gut medium containing 500 ng mL$^{-1}$ FGF10 and 1% FBS until day 17[27]. Medium was replaced every 3–4 days.

To generate intestinal organoids, MHG spheroids were cultured in 3D in basal gut medium containing 100 ng mL$^{-1}$ EGF, 100 ng mL$^{-1}$ NOGGIN, and 500 ng mL$^{-1}$ R-spondin1 until day 17. To generate colonic organoids, MHG spheroids were cultured in 3D in basal gut medium containing 100 ng mL$^{-1}$ EGF and 50 ng mL$^{-1}$ BMP4[26,30]. Medium was replaced every 3–4 days.

## Generation of H2B-mCherry H1 cell line

For lentivirus packaging, 6 μg of PLKO-H2B-mCherry overexpression plasmid and packaging plasmids were co-transfected into HEK293T cells with Lipofectamine 3000 reagent. The virus was collected at 48 h and 72 h post transfection. To promote transduction efficiency, 8 μg mL$^{-1}$ of polybrene (Sigma, Catalog # H9268) was applied. The stably transduced cells were further selected by puromycin (3 μg mL$^{-1}$) for 3 days.

## Scratch assay

After DE induction (on day 4), the cell monolayer was scratched using 10 μL pipette tip to lift the cell sheet along the path of scratch. In our scratch-reattach assay, after scratching, we gently put the culture plate back into the incubator, allowing the lifted cell sheet to curl back and reattach to the intact monolayer next to the scratched area in 24 h. In our scratch-clear assay, we immediately cleared the lifted cell sheet

after scratch, using an aspiration glass pipet under the microscope, leaving only cell monolayer next to the path of the scratch.

## Immunostaining

For cell monolayer, cells were fixed with 4% paraformaldehyde (PFA) for 30 min at room temperature before being permeabilized with 0.3% Triton X100 solution in PBS for another 30 min at room temperature. For spheroids and organoids, the samples were fixed with 4% PFA overnight at 4 °C before being treated with 0.3% Triton X100 for another overnight at 4 °C. For cryosectioning, 3D organoids were dehydrated in 30% sucrose solution in PBS overnight at 4 °C, then embedded in OCT compound and frozen. Samples were sectioned with the thickness of 15 μm. All samples were blocked with 5% donkey serum for 2 h before incubation with primary antibodies (the sources and dilutions were listed in Supplementary Table 1) for 2 days at 4 °C. Samples were then incubated with Alexa Fluor 488, Alexa Fluor 568, and/or Alexa Fluor 647-labeled, donkey-raised secondary antibodies (Thermo Fisher Scientific, 1:500 dilution) for another 2 days at 4 °C. In addition, FITC conjugated phalloidin (Thermo Fisher Scientific) was used for labeling F-actin, Alexa-Fluor dye-conjugated wheat germ agglutinin (WGA; Thermo Fisher Scientific) was used as a pan-cell membrane marker, and 4,6-diamidino-2-phenylindole (DAPI, Thermo Fisher Scientific) was used for labeling cell nuclei.

## Imaging and Image analysis

Immunostaining samples were imaged using Nikon-A1 and Zeiss laser scanning confocal microscope with control software including NIS-Elements AR 4.30.02 and ZEN 3.3 (Blue edition), respectively. Fluorescence images acquired from confocal microscopy were analyzed using NIS Viewer Ver 5.21.00 (Nikon) and ZEN blue edition (Zeiss), respectively, and reconstructed in 3D using Imaris8.2. To record gut tissue morphogenesis in μGSG, H2B-mCherry H1 cells were used and time-lapse images were acquired every 30 min for a total duration of 24 h while cell culture was maintained at 37 °C and 5% $CO_2$.

Using ImageJ 1.47, colony thickness was measured manually from confocal micrographs showing X-Z sections of cells colonies stained with WGA and DAPI. Fluorescence intensities were also measured from confocal micrographs using ImageJ along the path of interest.

## Analysis of spheroid morphology

Spheroid morphology was analyzed by manually tracing the spheroid boundary using ImageJ 1.47. The spheroid was fitted as an ellipse, with its minor axis length, aspect ratio, and projected area measured.

## Finite element analysis

The finite element simulations were performed using the commercial software Abaqus (Abaqus 6.14, Dassault Systèmes®). Based on the schematic of tissue architecture shown in Fig. 3n, we built a plane-stress model with Abaqus/Standard for buckling simulation. The model consisted of a 200-micrometers-long layered structure, in which the F-actin-rich cortex was simulated as a stiff film with a thickness of 2 micrometers while the underlying tissue was simulated as a compliant substrate with different thickness ranging from several micrometers to tens of micrometers[56]. The Young's modulus of the F-actin cortex and the tissue layer were set to be $10^4$ Pa and $10^1$ Pa[43]. Both the cortex and the tissue layer were assumed to be incompressible elastic matter. We introduced lateral thermal expansion to the tissue model, in order to mimic tissue crowding-induced internal compressive strain. The $y$-direction displacement at the lower boundary of the model was constrained, while the $x$-direction displacements at the left and right boundaries of the model were constrained to reflect tissue confinement. We used a uniform mesh grid (element size 0.2 micrometers) when the thickness of the tissue layer was smaller than 30 micrometers. A hybrid mesh grid consisting of uniform mesh grid (element size 0.2 micrometers) and a

heterogeneous grid (element size from 0.2 micrometers to 10 micrometers) when the thickness of the tissue layer was larger than 30 micrometers. The eight-node plane-stress biquadratic, reduced integration elements (CPS8R) were used to discretize both the cortex and the tissue layer. The convergence of all simulations was carefully validated to ensure that the results were mesh-independent.

## Active phase field model

We used an active phase field model to recapitulate the fission of gut spheroids observed in our experiment. We focused on a two-dimensional model for simplicity. The activity involved in our model refers to the spontaneous energy transformation from chemical energy to mechanical energy, which enables tissue movement and deformation. The tissue is described by a scalar field $\phi$, where $\phi = 1$ denotes the biological tissue, $\phi = -1$ is the surrounding environment, and $\phi = 0$ represents the boundary of the tissue. The diffusion kinetics of $\phi$ obeys

$$\partial_t \phi = M \nabla^2 \mu, \tag{1}$$

$$\mu = \frac{\delta F}{\delta \phi} + \lambda (\nabla \phi)^2, \tag{2}$$

where $M$ is the diffusion coefficient and $\mu$ is the chemical potential. The first term of the chemical potential Eq. (2) is determined by the variation of the free energy $F$,

$$F = \int \left[ -\frac{a}{2} \phi^2 + \frac{a}{4} \phi^4 + \frac{\kappa}{2} |\nabla \phi|^2 \right] d^2 r \tag{3}$$

The first two terms of the free energy function $F$ are Landau free energy, which is a double-well potential function whose minima are divided at $\phi = \pm 1$ to distinguish the two phases (i.e., tissue vs. medium). The parameter $a$ describes the magnitude of the Landau free energy, with greater $a$ to reflect a greater energy barrier between the two phases. The third term of $F$ is the surface energy to maintain the minimum surface area in the case of volume conservation, which reflects the active contraction behavior at the tissue surface. Parameter $\kappa$ is the surface tension coefficient that describes the magnitude of the surface energy and measures the strength of the contraction ability at the interface, which has been shown to be important in morphogenesis[45,46]. The second term of the chemical potential in Eq. (2) describes the active migration of $\phi$ (i.e., the change in the spatial distribution of cytoskeletal components for actomyosin contractility; reflected by active migration coefficient $\lambda$). In general, we assume that the active migration coefficient $\lambda < 0$, which leads to a lower chemical potential at the surface, and $\phi$ will migrate toward the surface, representing the directional accumulation of actomyosin machinery (e.g., F-actin and pMLC) at the outer tissue surface. The mathematical form of activity migration is derived from Activity Model B[44,48]. This enables our active phase field model to capture the accumulation of the actomyosin machinery and the resultant tension at the tissue surface during gut spheroid morphogenesis.

We normalize Eq. (1) by the unit length $L = 5\,\mu m$ and the unit time $T = L^2/Ma = 0.01$ hour in our simulation. The dimensionless surface tension and active migration coefficients become $K/(aL^2)$ and $\lambda/(aL^2)$, respectively. A double-period grid of size $200 \times 50$ with a space length of 0.5 is used. The initial conditions are set to $\phi = 1$ in a strip of width of 5 and $\phi = -1$ outside of this tissue column. To simulate the stochastic nature of tissue morphogenesis, a low-level white noise was also added.

## Inclusion and ethics

We declare no inclusion and ethics concerns related to this study.

## Statistics and reproducibility

Data analysis was performed using Microsoft Excel 2019 (Microsoft Corp.). Differences between groups were analyzed using GraphPad Prism8. No statistical method was used to predetermine sample size and the researchers who analyzed and counted results were blind to the conditions of the sample groups collected. Statistical analyses for comparisons of multiple samples were performed using a one-way analysis of variance (ANOVA) followed by an unpaired, two-sided Student's t-test. A $P < 0.05$ was considered statistically significant.

## Reporting summary

Further information on research design is available in the Nature Portfolio Reporting Summary linked to this article.

## Data availability

All data generated in this study are provided within the article and its supplementary information files. Source data are also provided with this paper. Source data are provided with this paper.

## Code availability

All custom codes used in this study have been deposited in the Github repository [https://github.com/Solongoodnight/Active-phase-field-model].

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

## Acknowledgements

Y.S.'s work is supported by the National Natural Science Foundation of China (U21A20203, 12102229), the National Key R&D Program of China (2022YFA1104600), the Oversea High-level Scholar Introduction Program, Tsinghua University Dushi Program, and the Tsinghua University Startup Funding. Y.S. also thanks the Major Basic Research Project of Science and Technology of Yunnan (202001BC070001 and 2019FY002) for supporting this work. S.S. is supported by the Mechanical Engineering Departmental Fellowship from the University of Michigan, Ann

Arbor. B.L. thanks the National Natural Science Foundation of China (11921002) for supporting this work. F.L. is supported by the National Natural Science Foundation of China (11902007) and the program of WIUCASQD2022023 from Wenzhou Institute, University of Chinese Academy of Sciences.

## Author contributions

F.L., X.L., S.S., and Y.S. conceived the project and designed experiments; F.L., X.L., S.S., J.B., Y.H., and L.S. performed experiments; Z.L., C.L., and B.L. developed the theoretical models for tissue morphogenesis; F.L., X.L., S.S., J.F., and Y.S. analyzed data and wrote the manuscript. Y.S. supervised the project. All authors contributed to the manuscript.

## Competing interests

The authors declare no competing interests.
