## [Peer Review File · Nature Communications]

Reviewers' comments:

Reviewer #1 (Remarks to the Author):

The authors report on a 2D micropatterned system (μ GSG) to generate gut spheroids from human pluripotent stem cells (hPSCs) through mechanically enhanced tissue morphogenesis. They report increases in gut spheroid yield independent of micropattern shape and size. An active phase field model is presented that supports an instability-driven mechanism driving spheroid morphogenesis in μ GSG.

Improving the yield and reproducibility of organoid technologies is an important need in the field. The manuscript is well written. The differentiation culture protocols and outcome metrics are well established in the field, so the contributions of the study lie on the use of the micropatterned substrates to improve organoid yield. As such, the scope of the study is very narrow. Furthermore, as detailed below, significant technical weaknesses limit the experimental aspects of the study. The active phase field model is an interesting aspect of the paper, but the validation of this model with experimental results is underdeveloped and the results are simply correlative. Taken together, the significance and innovation are relatively modest, and this manuscript may be more appropriate for a more specialized biophysics journal.

1. A critical flaw of the study is that spheroid yield results for the μ GSG system cannot be directly compared to the conventional culture method.
 - a. Initial cell densities were different (see point 2 below).
 - b. The density of Geltrex on the substrates may be different. The conventional tissue culture plate was coated with 1% Geltrex (according to manufacturer, 1 hr incubation at 37 deg C in Geltrex solution) whereas the micropatterned substrates were contact printed with PDMS stamps inked with 1% Geltrex and dried with nitrogen, printed on UV ozone-activated PDMS-coated glass slides, and exposed to 0.5% Pluronic F127. It is not clear that the same density/activity of Geltrex is obtained for the two approaches and the impact of Pluronic on Geltrex density or activity is not considered.
 - c. The timing for region-specific gut culture media including Y27632 concentration appears to be different between the traditional and μ GSG conditions.
2. The yield of gut spheroids is strongly dependent on the initial cell density (Fig. 3i). Importantly, the initial cell densities for conventional culture and μ GSG are different. The lower density used for the conventional culture (1.58×10^5 cells/cm²) can completely explain the reduced yield and apparent increases in spheroid yield for μ GSG. Indeed, when equivalent seeding densities were used between conventional culture and μ GSG (Fig. 3i) similar spheroid yields were obtained, and high spheroid yields were obtained for very large micropatterns (8000 μ m²) seeded with high cell densities. Therefore, the increase spheroid yield is mostly due to the higher cell seeding density and not the micropatterned substrates. This significantly reduces the significance and novelty of the study.
3. Although the contractility inhibitors Y27632 and blebbistatin are used to perturb spheroid formation (Fig. 4), these are not unexpected results. The mechanism responsible to the enhancement in spheroid production is superficially characterized.
4. Validation of active phase field model with experimental results is underdeveloped and the results are simply correlative. No direct experimental perturbation of the model parameters (accumulation of actomyosin machinery at the outer tissue surface or surface tension levels) is provided. In addition, how does the model incorporate cell density, which appears to drive the enhance in spheroid production?

Reviewer #2 (Remarks to the Author):

I read with pleasure the article "Mechanically enhanced biogenesis of gut spheroids with instability-driven morpho-mechanics", by F. Lin and co-workers. the article conveys two messages of equal relevance:

1. On the one hand, the authors develop a technology to produce gut spheroids with high throughput . This technology is based on the use of patterned surfaces as a substrate for cell culture. Such patterns effectively limit the available space and produce a 3D growth of cells, which eventually induces tubulation, pearling and fission.

Although not original, this approach is interesting and has been developed and described with care by the authors.

Nevertheless, I have few concerns about:

a) Why do the authors use patterns with a maximal surface of 8000 μm^2 ? The article reports that patterns increase the efficiency of spheroid production, as compared to non patterned surface. Interestingly, the efficiency does not depend on the pattern area, in the range between 50 and 8000 μm^2 . However, there must be a critical surface S^* , above which the efficiency progressively drops. This surface S^* would inform about the ability of a cell colony to feel the boundaries and to evolve collectively. Is that surface S^* comparable to that of an embryo, or to the size at which the gut start developing ?

I wish the authors to add such experiments to the revised article. Otherwise I will feel obliged to do the experiment myself, as I consider this point very relevant.

b) The authors should compare their approach with the other methods to produce spheroids (hanging drops, agarose cushions, 3D growth, ...). In the article, they insist on the great efficiency as compared to monolayer growth, which is known to be the most inefficient way to make spheroids.

c) Lines 71 and 72: "...unveils previously unappreciated mechanobiological paradigms for controlling tissue morphogenesis..." is an empty statement. Please, either remove it or specify which paradigm.

d) Line 81: "presumptive stomach region". What justify this presumption ?

e) Line 144: "in the scratch-clear assay". Please, describe how you immediately cleared the lifted cells after scratch (protocol).

2. On the other hand, the authors posit that tube fission is due to an active surface tension. They demonstrate experimentally that reducing the surface tension via a selective inhibition of acto-myosin activity reduces tube fission. The authors also propose a theoretical model to verify the hypothesis. I appreciate the idea, but I have some criticisms:

a) The model should be better explained. In particular, I do not understand the physical meaning of the term $a/2 \phi^2 + a/4 \phi^4$ in equation #3 (S.I.). This should be explicitly explained in the text.

b) The microscopic meaning of the parameter "a" should be discussed.

c) The theoretical model should be part of the main text, as this is the biological result that make the article suitable for Nature Communications.

d) The authors should discuss whether the model predicts mono-disperse spheroids, and if this prediction is compatible with the experimental results.

e) From the text, it is not clear if the model also predict the growth of a tubular structure or if the tube is taken as the initial (unexplained) state.

f) Line 181: "To test the above theory, we developed a morphomechanical model...". The authors probably meant "To test the above hypothesis, we developed a theoretical model..."

Yue Shao, Ph.D.

Associate Professor
Institute of Biomechanics and Medical Engineering
Department of Engineering Mechanics
School of Aerospace Engineering

Associate Director
Institute of Biomechanics and Medical Engineering

N423 Mong Man-Wei Science & Technology Building
30 Shuang Qing Road, Hai Dian District
Tsinghua University
Beijing, 100084

Office: 86-10-62782616
Mobile: 86-18110123091
yshao@tsinghua.edu.cn
<http://www.livingmachines.cn/>

Lin et al. “Mechanically enhanced biogenesis of gut spheroids with instability-driven morphomechanics”

Point-by-Point Response to Reviewers’ Comments

Reviewer #1

General Comments: *The authors report on a 2D micropatterned system (μ GSG) to generate gut spheroids from human pluripotent stem cells (hPSCs) through mechanically enhanced tissue morphogenesis. They report increases in gut spheroid yield independent of micropattern shape and size. An active phase field model is presented that supports an instability-driven mechanism driving spheroid morphogenesis in μ GSG.*

Improving the yield and reproducibility of organoid technologies is an important need in the field. The manuscript is well written. The differentiation culture protocols and outcome metrics are well established in the field, so the contributions of the study lie on the use of the micropatterned substrates to improve organoid yield. As such, the scope of the study is very narrow. Furthermore, as detailed below, significant technical weaknesses limit the experimental aspects of the study. The active phase field model is an interesting aspect of the paper, but the validation of this model with experimental results is underdeveloped and the results are simply correlative. Taken together, the significance and innovation are relatively modest, and this manuscript may be more appropriate for a more specialized biophysics journal.

Response: We thank this reviewer for recognizing the significance of improving the yield of organoid technology. We also thank this reviewer for appreciating the active phase field model that we developed. We also appreciate the reviewer’s insightful comments and useful advice, which are critical for us to strengthen the manuscript. After carefully reading all the comments, we noticed that some unclear descriptions in our previous manuscript might be the reason that causes some major concerns raised here. We believe further clarification below, along with additional data in our revision, could help address this reviewer’s concerns.

Specific Comment (1): *A critical flaw of the study is that spheroid yield results for the μ GSG system cannot be directly compared to the conventional culture method.*

Yue Shao, Ph.D.

Associate Professor
Institute of Biomechanics and Medical Engineering
Department of Engineering Mechanics
School of Aerospace Engineering

Associate Director
Institute of Biomechanics and Medical Engineering

N423 Mong Man-Wei Science & Technology Building
30 Shuang Qing Road, Hai Dian District
Tsinghua University
Beijing, 100084

Office: 86-10-62782616
Mobile: 86-18110123091
yshao@tsinghua.edu.cn
<http://www.livingmachines.cn/>

Response: We thank the reviewer for this critical comment. We agree that the comparison of spheroid generation efficiency between μ GSG-based and monolayer-based induction is a key result, which needs a scientific and reasonable comparison method.

For this reason, we specifically defined a method, from a biomanufacturing perspective, to quantify the spheroid generation efficiency, which is calculated as the ratio between the total number of spheroids generated on day 7 and the total number of initial hPSC input on day 0, *i.e.*, the “**input-output ratio**” of the spheroid biomanufacturing (initial hPSC number as the input, final spheroid number from the same amount of hPSC as output). This index has been commonly used as a measurement of the yield of biomanufacturing processes, and is a fair quantity to reflect how efficient the spheroid generation is, regardless of the culture systems used or the difference between the culture systems (*i.e.*, regardless of the culture systems, if one method allows us to obtain more spheroids than the other when using the same amount of total input hPSC, the method should be considered more efficient).

Given above rationale, despite differences in technical details between μ GSG and monolayer-based methods, above definition of spheroid generation efficiency should still allow us to make reasonable comparisons between μ GSG and traditional monolayer culture. Based on above method, our results and conclusions on the superior performance of μ GSG in generating gut spheroids over the traditional monolayer culture should stand justifiable and valid.

Of note, above definition of spheroid generation efficiency was exactly the same method that we used in the original manuscript. But we noted that our description of this method was unclear in our previous manuscript, and might cause confusions or misunderstandings, for which we sincerely apologize. Therefore, we further clarified above definition of methodology, as well as its rationale, in our revised manuscript and supplementary information.

Please see our revised manuscript (**Page 4, Line 85-88**) and supplementary information (**Page 6, Line 109-110**) for detailed information.

a. Initial cell densities were different (see point 2 below).

Yue Shao, Ph.D.

Associate Professor
Institute of Biomechanics and Medical Engineering
Department of Engineering Mechanics
School of Aerospace Engineering

Associate Director
Institute of Biomechanics and Medical Engineering

N423 Mong Man-Wei Science & Technology Building
30 Shuang Qing Road, Hai Dian District
Tsinghua University
Beijing, 100084

Office: 86-10-62782616
Mobile: 86-18110123091
yshao@tsinghua.edu.cn
<http://www.livingmachines.cn/>

Response to sub-point (a): We thank the reviewer for this comment. In fact, it has been reported by multiple previous studies that regardless how large, or small, the initial cell seeding density is used in traditional monolayer-based induction, it is difficult to generate gut spheroids efficiently (**Figure R1**, adapted from McCracken et al. 2011) (McCracken et al., 2011; Miller et al., 2019; Pitstick et al., 2022). Therefore, the main reason for the difference in spheroid generation efficiency between monolayer-induction and μ GSG-based induction was unlikely due to their difference in cell seeding density.

Figure R1. Spheroid generation from hPSC-derived definitive endoderm with different initial cell seeding densities – larger (1:3), optimal density (1:6) and small (1:12). Adapted from previous publication by McCracken et al. 2011.

In fact, the biggest difference between μ GSG and monolayer methods is that we replated DE cells on micropatterns on day 4, using μ GSG, which could result in cell rearrangement and increases the thickness of tissue; while monolayer method only cultures cells continuously until day 7, resulting in thinner tissue. This makes significant difference. We will further address this point in our **Response to Specific Comment (2)** below.

Yue Shao, Ph.D.

Associate Professor
Institute of Biomechanics and Medical Engineering
Department of Engineering Mechanics
School of Aerospace Engineering

Associate Director
Institute of Biomechanics and Medical Engineering

N423 Mong Man-Wei Science & Technology Building
30 Shuang Qing Road, Hai Dian District
Tsinghua University
Beijing, 100084

Office: 86-10-62782616
Mobile: 86-18110123091
yshao@tsinghua.edu.cn
<http://www.livingmachines.cn/>

b. The density of Geltrex on the substrates may be different. The conventional tissue culture plate was coated with 1% Geltrex (according to manufacturer, 1 hr incubation at 37 deg C in Geltrex solution) whereas the micropatterned substrates were contact printed with PDMS stamps inked with 1% Geltrex and dried with nitrogen, printed on UV ozone-activated PDMS-coated glass slides, and exposed to 0.5% Pluronics F127. It is not clear that the same density/activity of Geltrex is obtained for the two approaches and the impact of Pluronics on Geltrex density or activity is not considered.

Response to sub-point (b): We agree with the reviewer that it is important to investigate whether the superior spheroid generation by μ GSG is a property relied on its substrate preparation method. Indeed, monolayer-induction was conducted on substrates directly coated with 1% Geltrex for 1 h, while μ GSG-induction was performed on substrates made by microcontact printing (μ CP), which was done by a more sophisticated process. Substrates prepared by direct protein adsorption or μ CP might indeed have different surface conditions as the reviewer noted.

To investigate whether superior spheroid generation efficiency in μ GSG is dependent on the PDMS substrate prepared by μ CP method specifically, we also performed experiments with μ GSG using PDMS substrates prepared by direct coating of 1% Geltrex for 1 h. Notably, substrates prepared by μ CP or direct coating, despite their possible differences in surface Geltrex density or Pluronics, *etc.*, could both result in significantly enhanced spheroid generation with μ GSG (**Figure R2**), thereby suggesting the robustness of μ GSG on differently treated substrates.

Figure R2. μ GSG using substrates fabricated *via* microcontact printing (μ CP) or direct protein adsorption both showed enhanced spheroid generation

Yue Shao, Ph.D.

Associate Professor
Institute of Biomechanics and Medical Engineering
Department of Engineering Mechanics
School of Aerospace Engineering

Associate Director
Institute of Biomechanics and Medical Engineering

N423 Mong Man-Wei Science & Technology Building
30 Shuang Qing Road, Hai Dian District
Tsinghua University
Beijing, 100084

Office: 86-10-62782616
Mobile: 86-18110123091
yshao@tsinghua.edu.cn
<http://www.livingmachines.cn/>

We included these new data in our revised manuscript (**Page 5, Line 101-103**) and Supplementary Figure 5 (**Supplementary information; Page 17**), respectively.

c. The timing for region-specific gut culture media including Y27632 concentration appears to be different between the traditional and μ GSG conditions.

Response to sub-point (c): We thank this reviewer for pointing out this difference in the culture scheme between traditional and μ GSG culture. Indeed, we added Y27632 from day 4 to 5 in μ GSG culture (*i.e.*, from 0-24 h after we replated DE cells in μ GSG, to help ensure cell survival), but not in monolayer-based culture. To investigate whether treating monolayer-based culture with Y27632 from day 4 to 5 (*i.e.*, using the same timing and treatment scheme as used in μ GSG) could affect spheroid generation efficiency, we conducted new experiments and confirmed that even though treating monolayer culture with Y27632 the same as in the μ GSG culture, it did not improve spheroid formation (**Figure R3**).

Figure R3. Treating monolayer-based culture with the same chemical treatment scheme as in μ GSG (in particular, adding Y27632 from day 4-5) did not show notable improvement in spheroid generation

We included these new results in the revised manuscript (**Page 5, Line 98-100**) and Supplementary Figure 5 (**Supplementary information, Page 17**), respectively.

Specific Comment (2): The yield of gut spheroids is strongly dependent on the initial cell density (Fig. 3i). Importantly, the initial cell densities for conventional culture and μ GSG are

Yue Shao, Ph.D.

Associate Professor
Institute of Biomechanics and Medical Engineering
Department of Engineering Mechanics
School of Aerospace Engineering

Associate Director
Institute of Biomechanics and Medical Engineering

N423 Mong Man-Wei Science & Technology Building
30 Shuang Qing Road, Hai Dian District
Tsinghua University
Beijing, 100084

Office: 86-10-62782616
Mobile: 86-18110123091
yshao@tsinghua.edu.cn
<http://www.livingmachines.cn/>

different. The lower density used for the conventional culture (1.58×10^5 cells/cm²) can completely explain the reduced yield and apparent increases in spheroid yield for μ GSG. Indeed, when equivalent seeding densities were used between conventional culture and μ GSG (Fig. 3i) similar spheroid yields were obtained, and high spheroid yields were obtained for very large micropatterns ($8000 \mu\text{m}^2$) seeded with high cell densities. Therefore, the increase spheroid yield is mostly due to the higher cell seeding density and not the micropatterned substrates. This significantly reduces the significance and novelty of the study.

Response: We thank the reviewer for this critical comment. However, we want to point out that this reviewer's conclusion that "*when equivalent seeding densities were used between conventional culture and μ GSG (Fig. 3i) similar spheroid yields were obtained, and high spheroid yields were obtained for very large micropatterns ($8000 \mu\text{m}^2$) seeded with high cell densities. Therefore, the increase spheroid yield is mostly due to the higher cell seeding density and not the micropatterned substrates.*", was probably a misinterpretation of results shown in original Fig. 3a and Fig. 3i, which was likely due to the unclear presentation and labeling of the culture conditions in the original Fig. 3a and 3i of the previous manuscript. To address this point, we have made further clarifications in this revision, as elaborated below.

Firstly, we want to point out that it has been reported by multiple previous studies that regardless how large, or small, the initial cell seeding density is used in traditional monolayer-based induction, it is difficult to generate gut spheroids efficiently (McCracken et al., 2011; Miller et al., 2019; Pitstick et al., 2022). Therefore, the main reason for the difference in spheroid generation efficiency between monolayer-induction and μ GSG-based induction was unlikely due to their difference in cell seeding density.

Secondly, it is important to note that results in original Fig. 3a (where large $8000 \mu\text{m}^2$ micropatterns were used) and Fig. 3i (where different cell densities were tested) were ALL obtained from μ GSG; none of them was from conventional monolayer-based culture. In fact, it should be noted that the biggest difference between our μ GSG and traditional monolayer-based induction is that ours is a two-step method, and the traditional is a one-step method. Specifically, when we use μ GSG, we replated DE cells onto micropatterns at day 4, while

Yue Shao, Ph.D.

Associate Professor
Institute of Biomechanics and Medical Engineering
Department of Engineering Mechanics
School of Aerospace Engineering

Associate Director
Institute of Biomechanics and Medical Engineering

N423 Mong Man-Wei Science & Technology Building
30 Shuang Qing Road, Hai Dian District
Tsinghua University
Beijing, 100084

Office: 86-10-62782616
Mobile: 86-18110123091
yshao@tsinghua.edu.cn
<http://www.livingmachines.cn/>

traditional monolayer-induction does not have this step. Such operation results in more significant cell rearrangement and increases the thickness of the tissue in μ GSG. In contrast, monolayer-based induction shows much thinner tissue. This makes significant difference between the two methods.

Thus, in our original Fig. 3i, we changed the replating density of DE cells in μ GSG meant to control the thickness of the tissue layer to verify our hypothesis that it is tissue thickness, instead of micropattern's boundary or size, plays a key role in enhancing spheroid generation efficiency in μ GSG. And indeed, with results from above experiments and the scratch-reattach experiments, as well as new theoretical modeling results added in this revision (see our **Response to Specific Comment (3)** below), they support the notion that it is the tissue thickness and crowding, not cell seeding density, that plays the essential role in enhancing spheroid generation. But we also noted that our previous description about these results might be not clear enough, and we apologize for any confusion it caused.

Therefore, to avoid ambiguity and confusion, we added quantitative data of spheroid generation from monolayer-based induction (as a control) in our **revised Figure 3b&d**. We also clarified the description of our results in the revised manuscript (**Page 7, Line 147; Page 8, Line 162-167**).

***Specific Comment (3):** Although the contractility inhibitors Y27632 and blebbistatin are used to perturb spheroid formation (Fig. 4), these are not unexpected results. The mechanism responsible to the enhancement in spheroid production is superficially characterized.*

Response: We thank this reviewer for this constructive criticism. We also agree that the mechanism responsible to the enhancement in spheroid formation efficiency should be better elaborated.

To address this point, we revisited and re-analyzed our results shown in the original Fig. 3, and observed that there is a notable correlation between increased tissue thickness, as well as tissue crowding (*i.e.*, internal compressive deformation reflected by reduced size of nuclei), and the enhanced initiation of tissue buds from originally flat gut tissues in μ GSG from day 5-6 (**Figure R4**), which precedes the improved gut spheroid generation.

Yue Shao, Ph.D.

Associate Professor
 Institute of Biomechanics and Medical Engineering
 Department of Engineering Mechanics
 School of Aerospace Engineering

Associate Director
 Institute of Biomechanics and Medical Engineering

N423 Mong Man-Wei Science & Technology Building
 30 Shuang Qing Road, Hai Dian District
 Tsinghua University
 Beijing, 100084

Office: 86-10-62782616
 Mobile: 86-18110123091
 yshao@tsinghua.edu.cn
<http://www.livingmachines.cn/>

Figure R4. Cell multilayering and tissue crowding is necessary for enhancing the biogenesis of gut spheroids. (a) Confocal micrographs showing the X-Y and X-Z sections of gut tissue cultured in μ GSG under different cell plating density on day 5, with nuclei stained Blue. White arrowheads indicate cell multilayering. Purple dashed lines show the upper boundary of cell multilayering. Scale bar: 100 μ m. $n = 3$ independent experiments. (b) Scatter plot showing colony thickness in indicated culture conditions on day 5. (c) Scatter plot showing normalized number of spheroids generated by μ GSG under indicated conditions. (d) Scatter plot showing projected nucleus area in indicated conditions. (e) Scatter plot showing the correlation between tissue thickness and tissue crowding (*i.e.*, projected nucleus area). (f&g) Scatter plots showing the correlation between initiation of tissue buds and tissue thickness (f), and tissue crowding (g).

Given these observations, we hypothesize that μ GSG might leverage the synergy of increased tissue thickness and internal compressive deformation to enhance the efficiency of gut spheroid generation by facilitating the initiation of tissue buds from originally flat gut tissues *via* a buckling-like mechanism.

To examine this hypothesis, we developed a biomechanical model to recapitulate the architecture of gut tissues cultured in μ GSG, which features a thin, stiff F-actin-rich cortex atop a thick, soft tissue layer based on our experimental observation (**Figure R5a**). The

Yue Shao, Ph.D.

Associate Professor
 Institute of Biomechanics and Medical Engineering
 Department of Engineering Mechanics
 School of Aerospace Engineering

Associate Director
 Institute of Biomechanics and Medical Engineering

N423 Mong Man-Wei Science & Technology Building
 30 Shuang Qing Road, Hai Dian District
 Tsinghua University
 Beijing, 100084

Office: 86-10-62782616
 Mobile: 86-18110123091
 yshao@tsinghua.edu.cn
<http://www.livingmachines.cn/>

modulus of the F-actin-rich cortex and the underlying tissue bed were defined as 10^4 Pa and 10^1 Pa, respectively, based on previous measurements (Guo et al., 2017). Upon increasing internal compressive strain, this model could undergo buckling-like morphogenesis that resembles gut tissue budding from an otherwise flat tissue, as seen at day 5-6 (**Figure R5a**).

Specifically, using finite element method, we simulated tissue buckling behaviors driven by different levels of internal compressive strain in this model (**Figure R5b&c**). Our simulation results indicate that greater tissue thickness could facilitate the onset of tissue buckling under low-level tissue compression, and it further promotes tissue buckling amplitude under greater compressive strain. Together, these data substantiated that tissue multilayering and crowding might be an essential mechanism for enhancing gut spheroid formation in μ GSG *via* promoting tissue bud initiation through a buckling-like mechanism.

Figure R5. A biomechanical model recapitulates crowding guided tissue buckling to drive the initiation of tissue buds. (a) (*upper*) Representative X-Z section showing the architecture of the gut tissue cultured in μ GSG, with F-actin stained in green and nuclei in blue. The buckled configuration of F-actin cortex was outlined with the white dashed line. (*lower*) Schematic of the tissue architecture shown in the upper panel, which was used to conduct finite element analysis of tissue buckling under crowding-induced compressive strain. (b) Simulation results showing the change in normalized buckling amplitude as a function of the tissue-to-cortex thickness ratio (h_t/h_c). Results obtained under 1.2%, 2.5%, and 5% lateral compressive strain was shown, respectively. (c) Contours of tissue deformation under different levels of compressive strain for monolayer tissue ($h_t = 15\mu m$) and multilayer tissue ($h_t = 30\mu m$), respectively. Displacement field in the y direction (U_2) was color-coded as shown by the color bars.

Yue Shao, Ph.D.

Associate Professor
Institute of Biomechanics and Medical Engineering
Department of Engineering Mechanics
School of Aerospace Engineering

Associate Director
Institute of Biomechanics and Medical Engineering

N423 Mong Man-Wei Science & Technology Building
30 Shuang Qing Road, Hai Dian District
Tsinghua University
Beijing, 100084

Office: 86-10-62782616
Mobile: 86-18110123091
yshao@tsinghua.edu.cn
<http://www.livingmachines.cn/>

We incorporated above results in revised manuscript (**Page 8-9, Line 162-186**) and new Figure 3 (**Page 28**). We also added the method for the biomechanical model based on finite element method in our new supplementary information (**Page 9-10, Line 183-200**).

***Specific Comment (4):** Validation of active phase field model with experimental results is underdeveloped and the results are simply correlative. No direct experimental perturbation of the model parameters (accumulation of actomyosin machinery at the outer tissue surface or surface tension levels) is provided. In addition, how does the model incorporate cell density, which appears to drive the enhance in spheroid production?*

Response: We thank the reviewer for this critical comment. We agree that it is important to validate our active phase field model using experimental perturbations. In fact, after re-visiting our original manuscript, we noted that some key messages about our active phase field model and experimental validations were not clearly described and presented in the original manuscript, probably because we did not clearly distinguish results from “theoretical prediction” and “experimental validation” in the previous manuscript. Therefore, we extensively revised this part to provide better description of our active phase field model, its prediction, and how we experimentally validate multiple theoretical predictions.

Specifically, our active phase field model predicted two properties of gut spheroid fission, which were validated with experiments. (1) Our model predicted the mono-dispersed nature of spheroid morphogenesis, which was similarly seen in experiments, as shown in **revised Figure 5g and 5a-c**. (2) Our model, by tuning its two key parameters, namely, the surface tension coefficient and the active migration coefficient, predicted a biomechanical phase diagram of spheroid fission, as shown in **revised Figure 6a**; through experiments using small molecules disrupting either actin cytoskeleton or myosin-driven contractility, we manipulated the accumulation of actin and surface tension in our experiments, as shown in **revised Figure 6b&c**; and indeed, by quantitating tissue morphology under different treatment conditions, our results verified the mechano-sensitive fission-to-pearling transition predicted by the theoretical model, as shown in **revised Figure 6d&e**.

Yue Shao, Ph.D.

Associate Professor
Institute of Biomechanics and Medical Engineering
Department of Engineering Mechanics
School of Aerospace Engineering

Associate Director
Institute of Biomechanics and Medical Engineering

N423 Mong Man-Wei Science & Technology Building
30 Shuang Qing Road, Hai Dian District
Tsinghua University
Beijing, 100084

Office: 86-10-62782616
Mobile: 86-18110123091
yshao@tsinghua.edu.cn
<http://www.livingmachines.cn/>

In addition, we also agree with the reviewer that it is important to provide better explanation and understanding about why greater cell density (*i.e.*, larger tissue thickness) could enhance spheroid formation from an otherwise flat tissue. To address this point, we developed a new theoretical model, which supports our experimental observation that greater tissue thickness could facilitate the onset of tissue budding from an otherwise flat tissue *via* a buckling-like process, and further promote tissue budding/buckling amplitude under the condition of tissue crowding. This new model substantiated that tissue multilayering and crowding, as seen in experiments, might be an essential mechanism for enhancing gut spheroid formation in μ GSG *via* promoting tissue bud initiation through a buckling-like mechanism. Please see our **Response to Specific Comment (3)** above for details.

We have incorporated above revision and new data in the revised manuscript (**Page 10-13; Line 211-289**) and revised Figure 5&6 (**Page 33&35**), respectively.

Yue Shao, Ph.D.

Associate Professor
Institute of Biomechanics and Medical Engineering
Department of Engineering Mechanics
School of Aerospace Engineering

Associate Director
Institute of Biomechanics and Medical Engineering

N423 Mong Man-Wei Science & Technology Building
30 Shuang Qing Road, Hai Dian District
Tsinghua University
Beijing, 100084

Office: 86-10-62782616
Mobile: 86-18110123091
yshao@tsinghua.edu.cn
<http://www.livingmachines.cn/>

Reviewer #2

General Comment (1): *I read with pleasure the article “Mechanically enhanced biogenesis of gut spheroids with instability-driven morpho-mechanics”, by F. Lin and co-workers. the article conveys two messages of equal relevance:*

1. On the one hand, the authors develop a technology to produce gut spheroids with high throughput. This technology is based on the use of patterned surfaces as a substrate for cell culture. Such patterns effectively limit the available space and produce a 3D growth of cells, which eventually induces tubulation, pearling and fission.

Although not original, this approach is interesting and has been developed and described with care by the authors.

Response: We would like to thank the reviewer for this positive summary of our work, and these comments below are instrumental for us to strengthen our manuscript. We would like to thank the reviewer as well for taking the time to provide thoughtful feedback.

Nevertheless, I have few concerns about:

Specific Comment (1): *a) Why do the authors use patterns with a maximal surface of $8000 \mu\text{m}^2$? The article reports that patterns increase the efficiency of spheroid production, as compared to non patterned surface. Interestingly, the efficiency does not depend on the pattern area, in the range between 50 and $8000 \mu\text{m}^2$. However, there must be a critical surface S^* , above which the efficiency progressively drops. This surface S^* would inform about the ability of a cell colony to feel the boundaries and to evolve collectively. Is that surface S^* comparable to that of an embryo, or to the size at which the gut start developing?*

I wish the authors to add such experiments to the revised article. Otherwise I will feel obliged to do the experiment myself, as I consider this point very relevant.

Yue Shao, Ph.D.

Associate Professor
Institute of Biomechanics and Medical Engineering
Department of Engineering Mechanics
School of Aerospace Engineering

Associate Director
Institute of Biomechanics and Medical Engineering

N423 Mong Man-Wei Science & Technology Building
30 Shuang Qing Road, Hai Dian District
Tsinghua University
Beijing, 100084

Office: 86-10-62782616
Mobile: 86-18110123091
yshao@tsinghua.edu.cn
<http://www.livingmachines.cn/>

Response: We thank the reviewer for this comment. If we understand correctly, this reviewer seems to think that the enhanced spheroid generation efficiency in μ GSG must be caused by the boundary, or the edge-effect, of micropatterns, which represents a classical paradigm of how micropattern regulate tissue morphogenesis. However, this is not the case in our present work. **We want to clarify that in this study, we find that it is the tissue thickness, not tissue edge (or area), that plays a key role in enhancing spheroid yield in μ GSG.**

Therefore, there is indeed a lack of critical area S^* , above which the efficiency progressively drops. After carefully reading this reviewer's comments and re-visiting our original manuscript, we noted that some key points related to the mechanism for the enhanced spheroid formation in μ GSG were not clearly presented in the previous manuscript, which might cause some confusion. Therefore, we want to make further clarification as below.

Firstly, we want to emphasize that the biggest difference between our μ GSG and traditional monolayer-based induction is that ours is a two-step method, and the traditional is a one-step method. Specifically, when we use μ GSG, we replated DE cells onto micropatterns at day 4, while traditional monolayer-induction does not have this step. Such operation results in more significant cell rearrangement and increases the thickness of the tissue in μ GSG. In contrast, monolayer-based induction shows much thinner tissue (**Figure R6**). This makes significant difference between the two methods

Figure R6. Representative X-Z sections of staining of F-actin (green) and the nuclei (blue) in μ GSG (*left*) and monolayer (*right*) culture, showing different tissue thickness.

Given that, we further examined whether increased tissue thickness plays a key role in enhancing spheroid generation efficiency. By modulating cell seeding density in μ GSG (see **Figure R4a&b** above), or using a scratch assay in monolayer-based induction (please see our **revised Figure 4**), we used multiple ways to perturb the tissue thickness and proved that increased tissue thickness is crucial (both necessary and sufficient) for enhancing the spheroid generation efficiency. The multilayering in μ GSG also induced notable cellular

Yue Shao, Ph.D.

Associate Professor
Institute of Biomechanics and Medical Engineering
Department of Engineering Mechanics
School of Aerospace Engineering

Associate Director
Institute of Biomechanics and Medical Engineering

N423 Mong Man-Wei Science & Technology Building
30 Shuang Qing Road, Hai Dian District
Tsinghua University
Beijing, 100084

Office: 86-10-62782616
Mobile: 86-18110123091
yshao@tsinghua.edu.cn
<http://www.livingmachines.cn/>

crowding and intra-tissue compressive deformation, as reflected by reduced size of cell nuclei under higher plating density of DE cells at day 5 (see **Figure R4d** above). These data suggest that mechanically enforced cell multilayering and crowding is necessary for enhanced spheroid formation by μ GSG. Furthermore, we observed a notable correlation between increased tissue thickness, as well as tissue crowding (*i.e.*, internal compressive deformation reflected by reduced size of nuclei), and enhanced initiation of tissue buds from originally flat gut tissues in μ GSG from day 5-6 (see **Figure R4e-g** above), which precedes the improved gut spheroid generation.

Given these observations, we hypothesize that μ GSG might leverage the synergy of increased tissue thickness and internal compressive deformation to enhance the efficiency of gut spheroid generation by facilitating the initiation of tissue buds from originally flat gut tissues *via* a buckling-like mechanism. To examine this hypothesis, we developed a biomechanical model to recapitulate the architecture of gut tissues in μ GSG, which features a thin, stiff F-actin-rich cortex atop a thick, soft tissue layer based on our experimental observation (see **Figure R5** above; **Methods** in our revised manuscript). The modulus of the F-actin-rich cortex and the underlying tissue bed were defined as 10^4 Pa and 10^1 Pa, respectively, based on previous measurements (Guo et al., 2017). Upon increasing internal compressive deformation, this model could undergo buckling-like morphogenesis that resembles gut tissue budding seen at day 5-6 (see **Figure R5a** above; **Methods** in our revised manuscript). Specifically, using finite element method, we simulated tissue buckling behaviors driven by different levels of internal compressive strain in this model (see **Figure R5b&c** above; **Methods** in our revised manuscript). Our simulation results indicate that greater tissue thickness could facilitate the onset of tissue buckling under low-level tissue compression, and it further promotes tissue buckling amplitude under greater compressive strain. Together, these data substantiated that tissue multilayering and crowding might be an essential mechanism for enhancing gut spheroid formation in μ GSG *via* promoting tissue bud initiation through a buckling-like mechanism.

From these results, we can conclude that the boundary / edge of the micropattern, thereby the size of micropattern, does not play a significant role. Instead, abovementioned results support that tissue thickness-dependent, yet tissue shape-independent, paradigm of tissue

Yue Shao, Ph.D.

Associate Professor
Institute of Biomechanics and Medical Engineering
Department of Engineering Mechanics
School of Aerospace Engineering

Associate Director
Institute of Biomechanics and Medical Engineering

N423 Mong Man-Wei Science & Technology Building
30 Shuang Qing Road, Hai Dian District
Tsinghua University
Beijing, 100084

Office: 86-10-62782616
Mobile: 86-18110123091
yshao@tsinghua.edu.cn
<http://www.livingmachines.cn/>

morphogenesis is another novel finding in this study that challenges conventional thoughts on how tissue morphogenesis is regulated on micropatterns.

We have incorporated above revision and new data in the revised manuscript (**Page 8-9; Line 162-186**) and revised Figure 3 (**Page 28**), respectively.

***Specific Comment (2):** b) The authors should compare their approach with the other methods to produce spheroids (hanging drops, agarose cushions, 3D growth, ...). In the article, they insist on the great efficiency as compared to monolayer growth, which is known to be the most inefficient way to make spheroids.*

Response: We thank the reviewer for this comment. In fact, Reviewer #1 also pointed out that there is a lack of comparison between our method of generating gut spheroids and other methods for generating spheroids (such as hanging drop, agarose cushions, 3D culture, *etc.*). We agree that such comparison is critical, and want to take this chance to address this point below.

At present, there are two strategies to generate gut spheroid from hPSC, namely, (1) spontaneous spheroid morphogenesis (like that in traditional monolayer culture or in our μ GSG-based induction in this work); (2) re-aggregation of dissociated gut progenitor cells (which could be done using a variety of tools such as microwells, handdrops, 3D embedding, *etc.*). Admittedly, re-aggregation of dissociated gut progenitor cells is definitely the most efficient strategy in terms of generating spheroids (they are generally 100% efficient in generating spheroids, regardless of the cell type used). However, such re-aggregation strategy has a known weakness, which is the cell heterogeneity and incomplete epithelialization of the spheroids. The reason being that the differentiated gut progenitor cells cannot be absolutely pure, if one dissociates all of the differentiated cells and re-aggregates them, such spheroids contain only partial the gut progenitor cells and do not exhibit good markers for epithelialization. Therefore, spheroids generated using re-aggregation approach lack the biological fidelity demanded for organoid manufacturing, even though they could be generated with high efficiency.

Yue Shao, Ph.D.

Associate Professor
 Institute of Biomechanics and Medical Engineering
 Department of Engineering Mechanics
 School of Aerospace Engineering

Associate Director
 Institute of Biomechanics and Medical Engineering

N423 Mong Man-Wei Science & Technology Building
 30 Shuang Qing Road, Hai Dian District
 Tsinghua University
 Beijing, 100084

Office: 86-10-62782616
 Mobile: 86-18110123091
 yshao@tsinghua.edu.cn
<http://www.livingmachines.cn/>

To experimentally demonstrate above points, we compared the performance of μ GSG and that of forced cell aggregation in generating gut spheroids, using custom-made U-bottom microwell to enforce the aggregation of singly dissociated PFG cells on day 7. Notably, forced cellular re-aggregation only produced spheroids that heterogeneously expressed PFG markers such as SOX17 and ECAD (**Figure R7a**). In addition, extended culture of these spheroids till day 17 only yielded FGO with inconsistent expression of SOX2 and GATA4 (**Figure R7b**). In contrast, μ GSG-derived PFG and FGO exhibit uniform expression of corresponding markers (**Figure R7**). This is probably due to the self-selective nature of spontaneous gut spheroid morphogenesis from an otherwise heterogeneous gut cell population (Amack and Manning, 2012; Green, 2008; Maitre et al., 2012; Valet et al., 2022). Therefore, these data suggest that forced cellular aggregation, despite its merits in efficiency, only produces gut spheroids that are suboptimal in biological fidelity and developmental potential compared to that derived from μ GSG. This is the reason why in the past ten years, even though various methods of forced cell re-aggregation have been known, the spontaneous spheroid morphogenesis (instead of any kind of re-aggregation method) has remained the gold standard for generating gut spheroids from hPSC.

Figure R7. Limited biological fidelity of PFG and FGO generated by forced cell aggregation. (a) Confocal micrographs showing immunostaining of E-cadherin (ECAD) and SOX17 in PFG spheroids made by forced cell aggregation (*upper*), as well as ECAD, VIMENTIN, GATA4, and SOX17 in μ GSG-PFG spheroids (*lower*). DAPI stains the nuclei. $n = 3$

Yue Shao, Ph.D.

Associate Professor
Institute of Biomechanics and Medical Engineering
Department of Engineering Mechanics
School of Aerospace Engineering

Associate Director
Institute of Biomechanics and Medical Engineering

N423 Mong Man-Wei Science & Technology Building
30 Shuang Qing Road, Hai Dian District
Tsinghua University
Beijing, 100084

Office: 86-10-62782616
Mobile: 86-18110123091
yshao@tsinghua.edu.cn
<http://www.livingmachines.cn/>

independent experiments. The denominator reflects the total number of PFG spheroids quantitated, while the numerator is the number of indicated PFG phenotype among the quantitated PFG spheroids. Scale bar: 50 μm . (b) Confocal micrographs showing immunostaining of GATA4 and SOX2 in day 17 hFGO generated from PFG spheroids made by forced cell aggregation (*upper*), as well as in day 17 $\mu\text{GSG-PFG}$ (*lower*). DAPI stains the nuclei. $n = 3$ independent experiments. The denominator reflects the total number of hFGO quantitated, while the numerator is the number of indicated hFGO phenotype among the quantitated hFGO. Scale bars, 100 μm .

We have incorporated above revision and new data in the revised manuscript (**Page 6-7; Line 123-135**) and supplementary figure 10 (**Supplementary Information; Page 24**), respectively.

Specific Comment (3): *c) Lines 71 and 72: “...unveils previously unappreciated mechanobiological paradigms for controlling tissue morphogenesis...” is an empty statement. Please, either remove it or specify which paradigm.*

Response: We thank the reviewer for this critical comment. We have rephrased the description, which is now read as “*Notably, we demonstrate that μGSG -enhanced biogenesis of gut spheroids is independent of micropattern shape and size, but driven by mechanically enforced cell multilayering and crowding, suggesting a micropattern edge-insensitive, buckling-like mechanism to regulate initial tissue budding from originally flat gut tissues. We further combine experimental findings and an active-phase-field theory to recapitulate morphomechanics mechanism of subsequent spheroid pearling and fission in μGSG . This work provides an efficient, universal, scalable, and standardized system to produce human gut spheroids, and unveils mechanobiology paradigms based on tissue architecture and surface tension for controlling tissue morphogenesis*”. (**Page 4, Line 65-73**)

Specific Comment (4): *d) Line 81: “presumptive stomach region” . What justify this presumption ?*

Yue Shao, Ph.D.

Associate Professor
Institute of Biomechanics and Medical Engineering
Department of Engineering Mechanics
School of Aerospace Engineering

Associate Director
Institute of Biomechanics and Medical Engineering

N423 Mong Man-Wei Science & Technology Building
30 Shuang Qing Road, Hai Dian District
Tsinghua University
Beijing, 100084

Office: 86-10-62782616
Mobile: 86-18110123091
yshao@tsinghua.edu.cn
<http://www.livingmachines.cn/>

Response: By “presumptive stomach region”, we meant to describe the region (posterior foregut; PFG) that would further develop to become the stomach. This is a term commonly used in developmental biology. To identify such a presumptive stomach region, we used biological markers such as GATA4, SOX2, HNF1 β , FOXA2, and SOX17.

Specific Comment (5): *e) Line 144: “in the scratch-clear assay”. Please, describe how you immediately cleared the lifted cells after scratch (protocol).*

Response: We thank the reviewer for this critical comment. We have added the operational description, which now reads “*In our scratch-clear assay, we immediately cleared the lifted cell sheet after scratch, using an aspiration glass pipet under microscope, leaving only cell monolayer next to the path of the scratch.*”, in revised manuscript (**Page 9, Line 193-194**) and supplementary information (**Page 8, Line 152-153**).

General Comment (2): *On the other hand, the authors posit that tube fission is due to an active surface tension. They demonstrate experimentally that reducing the surface tension via a selective inhibition of acto-myosin activity reduces tube fission. The authors also propose a theoretical model to verify the hypothesis. I appreciate the idea, but I have some criticisms:*

Response: We thank the reviewer for this positive summary of our work. We would address each specific comment below.

Specific Comment (6): *a) The model should be better explained. In particular, I do not understand the physical meaning of the term $a/2 \phi^2 + a/4 \phi^4$ in equation #3 (S.I.). This should be explicitly explained in the text.*

Response: We thank the reviewer for pointing out this issue. We have specified the meaning of each term in this equation in our revised manuscript. The first two terms of the free energy function, *i.e.*, equation #3 (S.I.), are Landau free energy, which is a double-well potential function whose minima are divided at $\phi = \pm 1$ to distinguish the two phases, *i.e.*, the “tissue”

Yue Shao, Ph.D.

Associate Professor
Institute of Biomechanics and Medical Engineering
Department of Engineering Mechanics
School of Aerospace Engineering

Associate Director
Institute of Biomechanics and Medical Engineering

N423 Mong Man-Wei Science & Technology Building
30 Shuang Qing Road, Hai Dian District
Tsinghua University
Beijing, 100084

Office: 86-10-62782616
Mobile: 86-18110123091
yshao@tsinghua.edu.cn
<http://www.livingmachines.cn/>

phase and “medium” phase. The third term of F is the surface energy to maintain the minimum surface area in the case of volume conservation to describe the active contraction behavior at the interface. The parameter, a , describes the magnitude of the Landau free energy. The larger the a , the higher the energy barrier between the tissue phase and the medium phase. Parameter κ is the surface tension coefficient that describes the magnitude of the surface energy and measures the strength of the contraction ability at the interface. We also revised the text of our manuscript (**Page 11-12, Line 229-249**) and supplementary information (**Page 11, Line 213-219**) to better explain other aspects of our model.

Specific Comment (7): b) The microscopic meaning of the parameter “ a ” should be discussed.

Response: We thank the reviewer for pointing out this issue. Please see our **Response to Specific Comment (6)** above.

Specific Comment (8): c) The theoretical model should be part of the main text, as this is the biological result that make the article suitable for Nature Communications.

Response: We thank the reviewer for the approval of the novelty and significance of our model. We have included the development and explanations of this theoretical model in the main text of the revised manuscript (**Page 11-12, Line 229-249**).

Specific Comment (9): d) The authors should discuss whether the model predicts mono-disperse spheroids, and if this prediction is compatible with the experimental results.

Response: We thank the reviewer for this suggestion. Indeed, our model predicted the mono-dispersed spheroid morphogenesis, which is compatible with the experimental results. We included related discussion in the revised manuscript (**Page 12, Line 255-257**).

Yue Shao, Ph.D.

Associate Professor
Institute of Biomechanics and Medical Engineering
Department of Engineering Mechanics
School of Aerospace Engineering

Associate Director
Institute of Biomechanics and Medical Engineering

N423 Mong Man-Wei Science & Technology Building
30 Shuang Qing Road, Hai Dian District
Tsinghua University
Beijing, 100084

Office: 86-10-62782616
Mobile: 86-18110123091
yshao@tsinghua.edu.cn
<http://www.livingmachines.cn/>

Specific Comment (10): e) *From the text, it is not clear if the model also predict the growth of a tubular structure or if the tube is taken as the initial (unexplained) state.*

Response: In our active phase field model, the undeformed tubular structure was taken as an initial state. In order to recapitulate the initiation of tissue budding from an otherwise flat tissue, we developed a new model in this revision (see **Response to Specific Comment (1)** above) to illustrate how it occurs *via* a buckling-like process. In fact, further post-buckling deformation of the tissue could give rise to more extended finger-like structure emerging from the original flat tissue, and this could provide the basis for the initial tubular structure for simulating subsequent spheroid fission with the active phase field model. Therefore, with both theoretical models shown in the revised manuscript, it provides a more comprehensive explanation of the morphomechanics of gut spheroid formation.

Specific Comment (11): f) *Line 181: “To test the above theory, we developed a morphomechanical model...” . The authors probably meant “To test the above hypothesis, we developed a theoretical model...”*

Response: Response: We thank the reviewer for pointing out this issue. We have corrected the wording in the revised manuscript, which now reads “*To test above hypothesis, we developed a theoretical model from a reductionist viewpoint*” (**Page 10, Line 218**).

Yue Shao, Ph.D.

Associate Professor
Institute of Biomechanics and Medical Engineering
Department of Engineering Mechanics
School of Aerospace Engineering

Associate Director
Institute of Biomechanics and Medical Engineering

N423 Mong Man-Wei Science & Technology Building
30 Shuang Qing Road, Hai Dian District
Tsinghua University
Beijing, 100084

Office: 86-10-62782616
Mobile: 86-18110123091
yshao@tsinghua.edu.cn
<http://www.livingmachines.cn/>

References:

Amack, J.D., and Manning, M.L. (2012). Knowing the boundaries: extending the differential adhesion hypothesis in embryonic cell sorting. *Science* 338, 212-215.

Green, J.B. (2008). Sophistications of cell sorting. *Nat Cell Biol* 10, 375-377.

Guo, M., Pegoraro, A.F., Mao, A., Zhou, E.H., Arany, P.R., Han, Y., Burnette, D.T., Jensen, M.H., Kasza, K.E., Moore, J.R., *et al.* (2017). Cell volume change through water efflux impacts cell stiffness and stem cell fate. *P Natl Acad Sci USA* 114, E8618-E8627.

Maitre, J.L., Berthoumieux, H., Krens, S.F., Salbreux, G., Julicher, F., Paluch, E., and Heisenberg, C.P. (2012). Adhesion functions in cell sorting by mechanically coupling the cortices of adhering cells. *Science* 338, 253-256.

McCracken, K.W., Howell, J.C., Wells, J.M., and Spence, J.R. (2011). Generating human intestinal tissue from pluripotent stem cells in vitro. *Nature protocols* 6, 1920-1928.

Miller, A.J., Dye, B.R., Ferrer-Torres, D., Hill, D.R., Overeem, A.W., Shea, L.D., and Spence, J.R. (2019). Generation of lung organoids from human pluripotent stem cells in vitro. *Nature protocols* 14, 518-540.

Pitstick, A.L., Poling, H.M., Sundaram, N., Lewis, P.L., Kechele, D.O., Sanchez, J.G., Scott, M.A., Broda, T.R., Helmuth, M.A., Wells, J.M., *et al.* (2022). Aggregation of cryopreserved mid-hindgut endoderm for more reliable and reproducible hPSC-derived small intestinal organoid generation. *Stem Cell Rep* 17, 1889-1902.

Valet, M., Siggia, E.D., and Brivanlou, A.H. (2022). Mechanical regulation of early vertebrate embryogenesis. *Nature reviews Molecular cell biology* 23, 169-184.

REVIEWERS' COMMENTS

Reviewer #1 (Remarks to the Author):

The revised manuscript addressed several of my original technical concerns and the presentation is more clear. However, I remain unconvinced about the overall significance, impact, and novelty of the study. The work mostly presents an improved technique to produce gut spheroids but the relevance to the broad organoid community is modest. The model, although interesting, is correlative to experimental results and provides limited new insights. This manuscript is more appropriate for a more specialized biophysics journal.

Reviewer #2 (Remarks to the Author):

I have carefully read the new version of the manuscript and the rebuttal letter, in which the authors answer my and the other referee's comments.

The answers seem to me convincing and the changes made to the article are substantial. As far as I am concerned, all the doubts raised in my first review have been addressed.

I still think the authors focus too much on the "micropatterned gut spheroid generator (μ GSG)", in particular at the beginning of the article. As they explain in the letter, it is not micropatterning that helps spheroid formation, but the double seeding protocol. Thus, stressing so much on μ GSG throughout the first part of the article is misleading to the reader.

Yue Shao, Ph.D.

Associate Professor
Institute of Biomechanics and Medical Engineering
Department of Engineering Mechanics
School of Aerospace Engineering

Associate Director
Institute of Biomechanics and Medical Engineering

N423 Mong Man-Wei Science & Technology Building
30 Shuang Qing Road, Hai Dian District
Tsinghua University
Beijing, 100084

Office: 86-10-62782616
Mobile: 86-18110123091
yshao@tsinghua.edu.cn
<http://www.livingmachines.cn/>

Lin et al. “Mechanically enhanced biogenesis of gut spheroids with instability-driven morphomechanics”

Point-by-Point Response to Reviewers’ Comments

Reviewers' comments:

Reviewer #1

General Comments: The revised manuscript addressed several of my original technical concerns and the presentation is more clear. However, I remain unconvinced about the overall significance, impact, and novelty of the study. The work mostly presents an improved technique to produce gut spheroids but the relevance to the broad organoid community is modest. The model, although interesting, is correlative to experimental results and provides limited new insights. This manuscript is more appropriate for a more specialized biophysics journal.

Response: We thank this reviewer once again for all the insightful comments and suggestions, which have been critical for us to strengthen the manuscript.

Yue Shao, Ph.D.

Associate Professor
Institute of Biomechanics and Medical Engineering
Department of Engineering Mechanics
School of Aerospace Engineering

Associate Director
Institute of Biomechanics and Medical Engineering

N423 Mong Man-Wei Science & Technology Building
30 Shuang Qing Road, Hai Dian District
Tsinghua University
Beijing, 100084

Office: 86-10-62782616
Mobile: 86-18110123091
yshao@tsinghua.edu.cn
<http://www.livingmachines.cn/>

Reviewer #2

General Comment: *I have carefully read the new version of the manuscript and the rebuttal letter, in which the authors answer my and the other referee's comments.*

The answers seem to me convincing and the changes made to the article are substantial. As far as I am concerned, all the doubts raised in my first review have been addressed.

I still think the authors focus too much on the "micropatterned gut spheroid generator (μ GSG)", in particular at the beginning of the article. As they explain in the letter, it is not micropatterning that helps spheroid formation, but the double seeding protocol. Thus, stressing so much on μ GSG throughout the first part of the article is misleading to the reader.

Response: We thank this reviewer for previous helpful suggestions that guided us to improve our manuscript. We also agree with this reviewer that stressing too much on μ GSG throughout the first part of the article might be not helpful for the audience to grasp the key mechanism discovered subsequently in this study. Therefore, we reduced the emphasis on μ GSG, by reducing the use of this term where it is not necessary, in the first part of our main text. We also explicitly pointed out that μ GSG features a double cell-plating protocol that distinguishes it from traditional micropattern-based cultures. We have incorporated above changes in the revised manuscript.